biomechanics/mechanics

animal locomotion, centre-of-mass limits, minimal swim speed, great hammerhead, heterocercal tail

**Author for correspondence:**
Gil Iosilevskii
e-mail: igil@technion.ac.il

# Centre-of-mass and minimal speed limits of the great hammerhead

## Gil Iosilevskii

Faculty of Aerospace Engineering, Technion, Haifa, Israel

GI, 0000-0002-4114-3214

The great hammerhead is denser than water, and hence relies on hydrodynamic lift to compensate for its lack of buoyancy, and on hydrodynamic moment to compensate for a possible misalignment between centres of mass and buoyancy. Because hydrodynamic forces scale with the swimming speed squared, whereas buoyancy and gravity are independent of it, there is a critical speed below which the shark cannot generate enough lift to counteract gravity, and there are anterior and posterior centre-of-mass limits beyond which the shark cannot generate enough pitching moment to counteract the buoyancy–gravity couple. The speed and centre-of-mass limits were found from numerous wind-tunnel experiments on a scaled model of the shark. In particular, it was shown that the margin between the anterior and posterior centre-of-mass limits is a few tenths of the product between the length of the shark and the ratio between its weight in and out of water; a diminutive 1% body length. The paper presents the wind-tunnel experiments, and discusses the roles that the cephalofoil and the pectoral and caudal fins play in longitudinal balance of a shark.

## 1. Introduction

In order to swim along a straight path at constant speed and depth, the forces and moments acting on a shark should cancel out. A negatively buoyant shark will need hydrodynamic lift to cancel out the excess weight (the difference between gravity and buoyancy), thrust to cancel out the drag, and hydrodynamic pitching moment to cancel out the buoyancy–gravity couple (figure 1). Because all hydrodynamic forces scale with swimming speed squared, whereas gravity and buoyancy are independent of it, there is a critical speed below which a negatively buoyant shark will not be able to generate enough hydrodynamic lift to counteract the excess weight, and there are anterior and posterior limits on the centre-of-mass position beyond which the shark will not be able to generate enough hydrodynamic moment to counteract the hydrostatic couple. This study aims to find both

**Figure 1.** Balance of forces on a shark ascending at angle $\gamma$ relative to horizon; its angle of attack is $\alpha$ (it is measured between the caudo-cranial axis and the direction of swimming). $B$ and $G$ are the buoyancy and gravity respectively; $L$, $D$ and $T$ are the lift, drag and thrust; $M_y$ is the hydrodynamic pitching moment about the centre of buoyancy. $C_L$, $C_D$, $C_T$ and $C_M$ are the respective dimensionless coefficients. $q$, $S$ and $l$ are the dynamic pressure $(1/2)\rho v^2$, maximal cross-section area of the body and the fork length. Subscripts 'cf' and 'no cf' mark the respective contributions of the caudal fin and of the rest of the shark. $\lambda_{cf}$ is the ratio between lift and thrust of the caudal fin. Centre of buoyancy is located $0.455l$ posterior to the snout and $0.0122l$ dorsal to the caudo-cranial axis; centre of pressure of the caudal fin is assumed located $0.685l$ posterior to the centre of buoyancy and $0.04l$ dorsal to it. The distance between centres of mass (cm) and buoyancy (cb) has been exaggerated for clarity. $x$ and $z$ are the axes of $R^B$; $x^E$ and $z^E$ are the axes of $R^E$ (the origin of both frames is arbitrary). $x$-axis coincides with the caudo-cranial axis of the body, $x^E$-axis is horizontal.

the minimal speed and the centre-of-mass limits of the great hammerhead shark *Sphyrna mokarran* (Ruppell), but the quest for these limits poses a much broader question on the roles of the cephalofoil and the pectoral and caudal fins in longitudinal balance of the shark [1–5].

Anterior and posterior limits on the centre-of-mass position depend on the hydrodynamic pitching moment a shark can generate, which, in turn, depends on distribution of hydrodynamic forces along the shark. This distribution changes with the angle between the caudo-cranial axis of the body and the swimming direction (aka the angle of attack, $\alpha$; figure 1 and table 1), alignment angles of the pectoral fins and of the cephalofoil relative to the body ($\delta_{pf}$ and $\delta_c$, respectively), and the part that the caudal fin takes in generation of lift. By changing the lift of the caudal fin, the shark may alter its balance in the same way it can alter it by changing the alignment angles of the fins. We do not know *a priori* what part the caudal fin really takes in generation of lift, and we can only guess that a shark has control over it. Nonetheless, being generated by the same pressure distribution, lift $L_{cf}$ and thrust $T_{cf}$ of the caudal fin should be comparable quantities. In fact, their ratio $\lambda_{cf} = L_{cf}/T_{cf}$ in leopard and bamboo sharks is known to be almost unity [2] (this conjecture was inferred from the angle between the direction of flow in the wake of the shark and the direction of swimming). In what follows, $\lambda_{cf}$ will replace $L_{cf}$ and an independent variable. For the sake of definiteness, it will be assumed bounded to interval (0,1), but as long as it remains of the order of unity, the particular range of this parameter is inconsequential to conclusions of this study.

As other ground sharks, the great hammerhead swims with subcarangiform gait, generating thrust both with its body and the caudal fin. Most of the thrust, however, can be associated with the part of the shark combining the largest dorso-ventral dimension with the largest lateral displacement during a tail-beat—i.e. the caudal fin [6,7]. Thrust *per se* is hardly consequential for the present analysis, but the lift is. Decreasing the share of the caudal fin in generation of thrust proportionally decreases its share in generation of lift (for the same $\lambda_{cf}$); in turn, a decrease in lift of the caudal fin moves the posterior limit of the centre of mass anteriorly. This effect can be accounted for by decreasing the viable range of $\lambda_{cf}$, but, as already mentioned, conclusions of this study should not be affected by it. We proceed under the assumption that thrust is generated solely by the caudal fin.

Thrust and drag are both defined as the components of hydrodynamic force in the direction of swimming, the former along it, and the latter opposing it. For a self-propelling body—as a swimming shark is—separation between the two is essentially impossible [8]. Consistent, however, with our associating the entire thrust with the respective component of the hydrodynamic force acting on the caudal fin (and the caudal fin only), we associate drag with the respective component of the hydrodynamic force acting on the rest of the shark as if it were moving stretched at the same speed

**Table 1.** Nomenclature.

| | |
|---|---|
| $B$ | buoyancy |
| $C_D$, $C_L$, $C_M$ | drag, lift and pitching moment coefficients: $C_L = L/qS$, $C_D = D/qS$, $C_M = M_y/qlS$ |
| $D$ | drag |
| $\widehat{Fr}$ | scaled speed (scaled Froude number): $v/\sqrt{g\beta l}$ |
| $G$ | gravity force: $G = mg$ |
| $g$ | acceleration of gravity |
| $k_{pc}$ | prismatic coefficient: the ratio between the volume of the shark and the minimal cylinder enclosing its body (with no fins) |
| $L$ | lift |
| $l$ | fork length |
| $M_y$ | pitching moment about the centre of buoyancy |
| $m$ | body mass: $m = \rho V/(1-\beta)$ |
| $q$ | dynamic pressure: $(1/2)\rho v^2$ |
| $R^B$, $R^S$, $R^E$ | reference frames |
| $S$ | maximal cross- (transverse-) section area of the body |
| $T$ | thrust: $T = qSC_T$ |
| $V$ | body volume: $V = k_{pc}Sl$ |
| $v$ | swim speed (shark) or air speed (wind-tunnel model) |
| $\widehat{X}$ | scaled horizontal margin between centres of mass and buoyancy: $(x_{cm}^E - x_{cb}^E)/(\beta l)$ |
| $x_{cm}$, $z_{cm}$ | location of the centre of mass in $R^B$ (caudo-cranial and dorso-ventral) |
| $x_{cm}^E$, $z_{cm}^E$ | same, but in $R^E$ (horizontal and vertical) |
| $x_{cb}$, $z_{cb}$ | location of the centre of buoyancy in $R^B$ (caudo-cranial and dorso-ventral) |
| $x_{cb}^E$, $z_{cb}^E$ | same, but in $R^E$ (horizontal and vertical) |
| $\alpha$ | angle of attack, the angle between the caudo-cranial axis and the swimming direction |
| $\beta$ | ratio of weights in and out of water |
| $\gamma$ | trajectory angle, the angle between the swimming direction and the horizon |
| $\delta_c$, $\delta_{pf}$ | set angles of the cephalofoil and the pectoral fins relative to the caudo-cranial axis |
| $\lambda_{cf}$ | ratio between lift and thrust of the caudal fin |
| $\rho$ | water or air density |
| *modifiers* | |
| $\cdots_c$ | associated with the cephalofoil |
| $\cdots_{cf}$ | (with or without a leading comma) associated with the caudal fin |
| $\cdots_{cp}$ | associated with the centre of pressure |
| $\cdots_{pf}$ | (with or without a leading comma) associated with the pectoral fins |
| $\cdots_{,no\ cf}$ | associated with the entire shark without the caudal fin |
| $\cdots^E$ | relative to $R^E$ |
| $\overline{\cdots}$ | reduced quantity with the fork length $l$ serving as a unit of length |
| $\widehat{\cdots}$ | reduced quantity with the product $\beta l$ serving as a unit of length |

and the same body angle. By doing so, we effectively separate hydrodynamic forces acting on the caudal fin from hydrodynamic forces acting on the rest of the shark, and open up the option to use wind-tunnel experiments to find the latter [1,9].

From hydrodynamic perspective, there is no difference between swimming in (practically incompressible) water and flying at low subsonic speeds in air as long as the respective Reynolds numbers are similar, and no cavitation occurs in water. Cavitation is not expected at swimming

speeds of a few metres per second which are relevant to this study [10], and hence forces measured on a model shark in a wind tunnel at low subsonic speeds (approx. $50 \, \text{m s}^{-1}$) can be straightforwardly re-scaled to find the forces that would have acted on the full-sized shark swimming with the same $\alpha$, $\delta_{pf}$ and $\delta_c$, at any speed.[1] In turn, knowing the forces acting on the shark as functions of speed, one could find the centre-of-mass position, the swimming speed and the thrust that would have balanced the shark swimming along a straight path for a given lift-to-thrust ratio of the caudal fin $\lambda_{cf}$. The lowest speed at which one could balance the shark for all viable combinations of $\alpha$, $\delta_{pf}$ and $\delta_c$, would be the minimal swim speed; at every swim speed above the minimal speed, the anterior and posterior extrema with respect to $\delta_c$ and $\delta_{pf}$ of the centre-of-mass positions that balances the shark would be the respective centre-of-mass limits at that speed. Centre-of-mass position that may allow the shark to swim at any speed above the minimal speed should be bounded between the most posterior of its anterior (speed-dependent) limits, and the most anterior of its posterior (speed-dependent) limits.

# 2. Material and methods

## 2.1. Model shark

The model of the shark (figure 2; electronic supplementary material, figures S1 and S2) was designed using CAD software (SolidWorks® 2014) based on available statistical data [11] and numerous photographs; it was printed in FullCure720. The model had replaceable fins, head and neck: different neck pieces allowed changing the angle of the cephalofoil relative to the body ($\delta_c$) between $-10°$ and $+10°$ in steps of $5°$; different pectoral fins allowed changing their angle relative to the body ($\delta_{pf}$) in the same range (figure 2 and electronic supplementary material, figure S2). The cephalofoil and all the fins had NACA 0015 profile.[2] The fork length ($l$) was 500 mm; the part of the model that went into the tunnel was 431 mm long, ending at the caudal end of the second dorsal fin (electronic supplementary material, figure S1). Its maximal cross-section area $S$ was $3870 \, \text{mm}^2$ ($0.0155 l^2$); its volume $V$ was $1\,236\,616 \, \text{mm}^3$ ($0.01 l^3$). The centre of buoyancy of the model shark (with the caudal fin) was 227.5 mm ($0.455 l$) posterior to the snout and 6.1 mm ($0.0122 l$) ventral to the caudo-cranial axis. Printer-ready STL files of the model can be found on the Dryad Digital Repository.

## 2.2. Reference frames

When addressing a shark that swims upright at constant (tail-beat averaged) speed along a straight path, three right-handed reference frames that follow the shark naturally come into use. The first one, $R^B$, has its $x$- and $z$-axes in the sagittal plane of its un-deformed body: the $x$-axis points anteriorly along the caudo-cranial axis, and the $z$-axis points ventrally (figure 1). Angle of attack $\alpha$ is measured between the $x$-axis of this frame and the swimming direction. The second frame, $R^S$, is rotated relative to $R^B$ about the $y$-axis through angle $-\alpha$, and hence its $x$-axis points in the swimming direction. Drag $D$ and lift $L$ are defined as the components of the hydrodynamic force along the negative directions of the $x$- and $z$-axes of this frame (figure 1). Thrust $T$ is defined as the component of force opposing drag. Swim-path angle $\gamma$ is measured between the $x$-axis of $R^S$ and the horizon (figure 1). The third frame, $R^E$, is rotated relative to $R^B$ about the $y$-axis through angle $-(\alpha + \gamma)$, and has its $x$-axis horizontal (figure 1). Buoyancy $B$ and gravity $G$ act along the $z$-axis of this frame, the first one in the negative direction (upwards); the second one in the positive direction (downwards). Pitching moment $M_y$ is defined as the $y$-component of the hydrodynamic moment, and it is the same in all three reference frames.

---

[1] In principle, yawing motion of a swimming shark may augment lift of the cephalofoil and pectoral fins, and make the forces reconstructed from wind-tunnel measurements underestimate the true forces. The augmentation factor should be a fraction of the ratio between the mean-square forward velocity of the distal margin of the respective lifting surface, $u$, and the swimming speed, $v$. Assuming sinusoidal yawing motion with frequency $f$ and angular amplitude $\theta_0$, $\langle u^2 \rangle / v^2 = 2(\pi f b \theta_0 / v)^2$, where $b$ is the distance between that margin and the sagittal plane. By interpretation, $b\theta_0$ is its anterior–posterior displacement, whereas the ratio $v/f$ is the stride length. The ratio of the two is invariably small, and so is $\langle u^2 \rangle / v^2$. A 3 m shark, which probably had the distal margin of its pectoral fins situated approximately 0.45 m from the sagittal plane, was logged swimming at $0.8 \, \text{m s}^{-1}$ with tail-beat frequency of 0.4 Hz [9]. Assuming $\theta_0 \sim 0.1$ rad (see supplementary video to [9]), $\langle u^2 \rangle / v^2 \sim 0.01$.

[2] There are no systematic data on profiles of the pectoral fins and the cephalofoil of sharks in general, and the great hammerhead in particular. At chord-based Reynolds numbers between 100 000 and 150 000, which are relevant to this study, a ubiquitous NACA 0015 performs practically as well as any symmetrical aerofoil with thickness ratio between 9 and 18% [12–14] and is thick enough to make three-dimensional printing in FullCure720 worries-free.

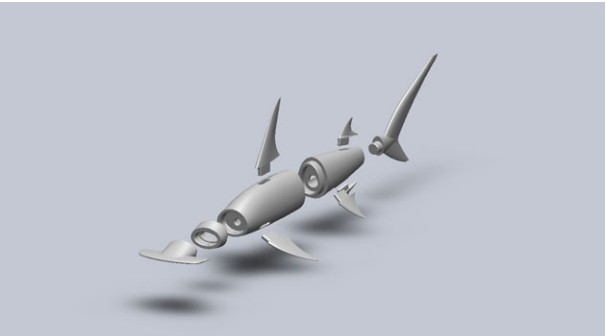

**Figure 2.** The CAD model of the shark. Different neck pieces (electronic supplementary material, figure S2) allowed changing the angle of the cephalofoil relative to the body between −10° and +10°. Different pectoral fins allowed changing their orientation relative to the body in the same range. The part of the model that was used for testing did not include the caudal fin. The balance was attached to an anchor implanted in the central piece; circular cut-off for the sting is visible in the posterior piece.

## 2.3. Collecting the data

The experiments were carried out at Tunnel 14 of the Faculty of Aerospace Engineering, Technion. This tunnel is of an open type, with $1 \times 1 \times 3$ m test section (electronic supplementary material, figure S3). In all the experiments presented here the air speed was $50$ m s$^{-1}$. At this air speed, the Reynolds number of the model shark was approximately the same as that of a 2.5 m (fork length) shark swimming at $0.7$ m s$^{-1}$ in 20°C water; turbulence intensity across the test section was approximately 0.2%. In a typical experiment, the model was assembled with the cephalofoil and the pectoral fins at preset angles relative to the body, and the angle of attack was changed continuously between −15° and 17°. The blockage ratio never exceeded 2%.

Forces and moments acting on the model were picked up by an in-house six-component sting balance (designated 646301 N/6461-3). The sting was 16 mm in diameter; its length allowed for a 255 mm gap ($0.6l$) between the caudal end of the model and the downstream attachment point of the sting. Measurement resolution was $1$ µV, which is equivalent to approximately $4.2$ mN of lift, $2.6$ mN of drag and $0.21$ mN m of pitching moment; measurement accuracy was approximately $10$ µV for all channels. Lift and drag during the experiments were of the order of 10 and 1 N, respectively; pitching moment was of the order of 2 N m. The data were acquired at 5 kHz. It was low-pass filtered at 4 Hz, and block averaged with 500 samples per block. Below stall, all experiments were repeatable to within the measurement accuracy.

## 2.4. Handling the data

Lift, drag and pitching moment acting on the shark were scaled with

$$L_{\text{no cf}} = qSC_{L,\text{no cf}}, \quad D_{\text{no cf}} = qSC_{D,\text{no cf}}, \quad M_{y,\text{no cf}} = qSlC_{M,\text{no cf}}, \tag{2.1}$$

where $C_{L,\text{no cf}}$, $C_{D,\text{no cf}}$ and $C_{M,\text{no cf}}$ are the respective dimensionless coefficients, whereas $q = (1/2)\rho v^2$, $S \approx 0.0155 l^2$ and $l$ are the dynamic pressure, maximal cross-section area and the fork length, respectively. When scaling the wind-tunnel experiments, $\rho$ and $v$ were the density of air and the air speed, respectively; when rescaling the results back to a swimming shark, they were the density of water and the swim speed. When presenting the wind-tunnel data, the pitching moment was invariably referred to the centre of buoyancy, $0.455l$ posterior to the snout and $0.0122l$ dorsal to the caudo-cranial axis (figure 1). The point along the caudo-cranial axis where the pitching moment vanishes,

$$x_{\text{cp,no cf}} - x_{\text{cb}} = l \frac{C_{M,\text{no cf}}}{C_{L,\text{no cf}} \cos \alpha + C_{D,\text{no cf}} \sin \alpha}, \tag{2.2}$$

is (somewhat loosely) referred to below as the 'centre of pressure'.

## 2.5. Reconstructing the forces on a swimming shark

Forces acting on a shark were assembled from hydrodynamic forces acting on its body and fins (with no caudal fin), hydrodynamic forces generated by the caudal fin, buoyancy and gravity. Hydrodynamic forces acting on the body of the shark were reconstructed from the wind-tunnel data using (2.1). Buoyancy $B = \rho g V$ was found from the scaled volume of the shark $V \approx 0.01 l^3$ ($g$ stands for the acceleration of gravity). Gravity $G = B/(1 - \beta)$ was found from buoyancy based on ratio between weights of the shark in and out of water, $\beta$. The fork length $l$ and the weights ratio $\beta$ were left unassigned because the results could be scaled using their product as a unit of length (see below). Thrust of the caudal fin $T_{cf}$ was found as the force needed to balance the forces on the shark in the swimming direction. Lift of the caudal fin was assumed as a product $L_{cf} = \lambda_{cf} T_{cf}$, where $\lambda_{cf}$ takes on any value in the interval (0,1). Both forces were assumed to act at $(x_{cf}, 0, z_{cf})$, $0.685l$ posterior to the centre of buoyancy (midway between the fork and the caudal end of the dorsal lobe) and $0.04l$ dorsal to it (figure 1). Changing the centre of action in the $0.04l$ vicinity of the chosen location yielded essentially the same results.

## 2.6. Balancing the shark

The shark was balanced in an upright posture swimming with a certain $\lambda_{cf} \in (0, 1)$ at constant speed along a straight path, inclined at angle $\gamma$ relative to horizon (figure 1). It was done for any (viable) combination of $\alpha$, $\delta_c$ and $\delta_{pf}$ by adjusting the swim speed, thrust and the distance between centres of mass and buoyancy (rather, the projection of this distance onto the horizontal plane, $x_{cm}^E - x_{cb}^E$). Details can be found in appendix A. The practical outcome was the scaled swim speed, $\widehat{\mathrm{Fr}} = v/\sqrt{g\beta l}$,

$$\widehat{\mathrm{Fr}} = \sqrt{2k_{pc}\frac{\cos\gamma - \lambda_{cf}\sin\gamma}{C_{L,\text{no cf}} + \lambda_{cf}C_{D,\text{no cf}}}}, \tag{2.3}$$

and the scaled horizontal margin between centres of mass and buoyancy, $\widehat{X} = (x_{cm}^E - x_{cb}^E)/(\beta l)$,

$$\widehat{X} = \frac{C_{M,\text{no cf}}(\cos\gamma - \lambda_{cf}\sin\gamma)}{C_{L,\text{no cf}} + \lambda_{cf}C_{D,\text{no cf}}} - \frac{C_{D,\text{no cf}}\cos\gamma + C_{L,\text{no cf}}\sin\gamma}{C_{L,\text{no cf}} + \lambda_{cf}C_{D,\text{no cf}}}$$
$$\times ((\lambda_{cf}\cos\alpha - \sin\alpha)(\bar{x}_{cb} - \bar{x}_{cf}) + (\lambda_{cf}\sin\alpha + \cos\alpha)(\bar{z}_{cb} - \bar{z}_{cf})), \tag{2.4}$$

(these are equations (A 15) and (A 16) in the appendix), which could be found directly from the wind-tunnel data. When the swim path is horizontal, $\widehat{X}$ practically becomes the scaled margin between the centre of pressure of the shark and the centre of buoyancy (see equation (4.2) below). Noting the common definition of the Froude number, $\mathrm{Fr} = v/\sqrt{gl}$, the scaled swim speed, $\widehat{\mathrm{Fr}}$, can be interpreted as its modified, buoyancy-corrected, variant.

# 3. Results

## 3.1. Wind-tunnel data

A representative set of wind-tunnel results can be found in figure 3, where lift, drag and pitching moment coefficients are displayed as functions of the angle of attack $\alpha$ for five angles of the cephalofoil $\delta_c$ and a single setting of the pectoral fins $\delta_{pf} = 0$. In general, both the lift and the pitching moment coefficients increase with the angle of attack and with the angle of the cephalofoil relative to the body (figure 3a,c), but the lift coefficient drops when the angle of attack exceeds 13°. At this angle, the pectoral fins stall, resulting in a loss of lift and an increase in drag (figure 3a,b). The cephalofoil stalls when its angle relative to the flow, $\alpha + \delta_c$, exceeds approximately 12°; in fact, when set at $\delta_c = 10°$, it stalls when $\alpha \approx 2°$. As a consequence, all configurations with $\delta_c \geq 0$ converge to the same lift and pitching moment coefficients at high angles of attack (figure 3a,c). The cephalofoil is responsible for up to one-third of the lift generated by the shark. Removing it (by replacing the head of the shark with the one resembling the head of a typical requiem shark—see electronic supplementary material, figures S1, S2) decreases both the lift slope and the maximal lift by approximately the same factor (figures 3a and 4a).

The centre of pressure (its location was computed using (2.2)) moves anteriorly with increasing angle of the cephalofoil relative to the body (figure 3d); it moves toward the anterior margin of the pectoral fins

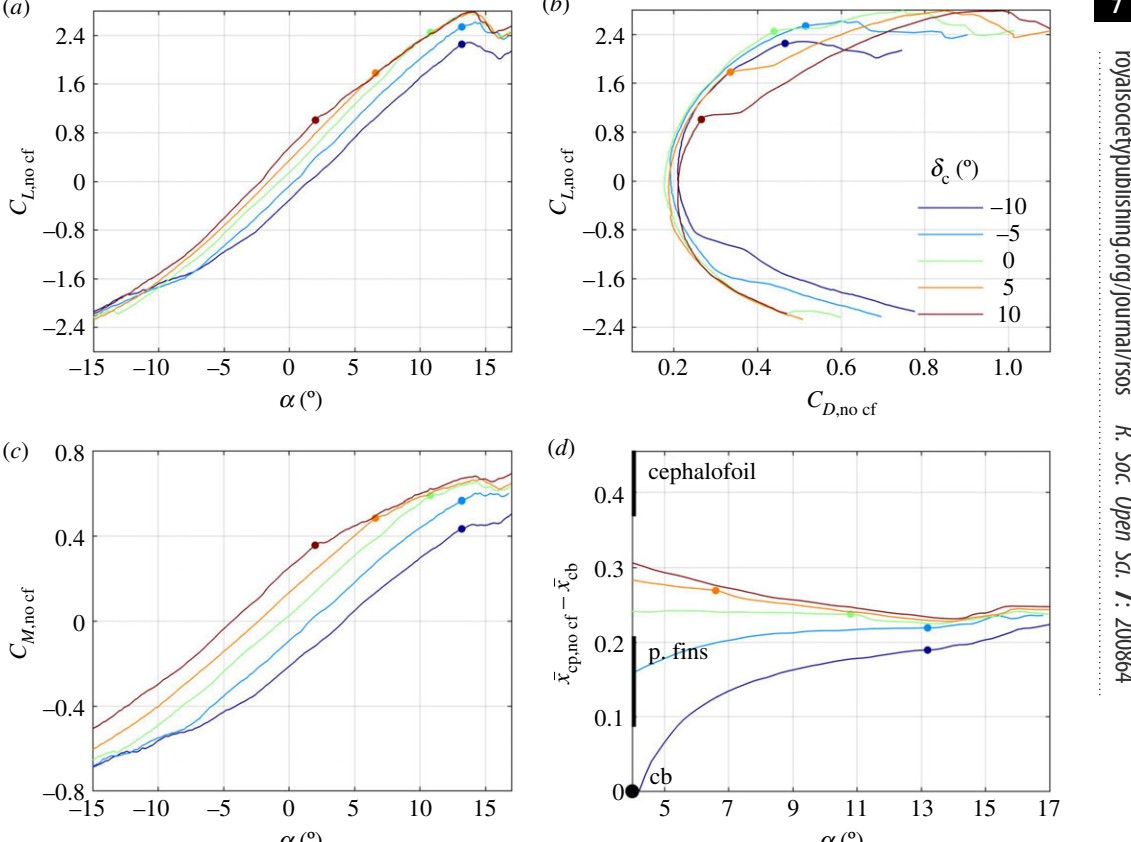

**Figure 3.** Lift coefficient (a), pitching moment coefficient (c) and the centre of pressure (d) as functions of angle of attack for five different orientations of the cephalofoil relative to the body (colour-coded as specified on plate (b)). The respective drag polars are shown on plate (b). In this set of experiments, the pectoral fins were aligned with the caudo-cranial axis ($\delta_{pf} = 0$). Small dots mark an estimated onset of the dorsal-side stall of the cephalofoil or the pectoral fins (it is the cephalofoil when $\delta_c \geq 0$). All surfaces stall when set at 11° to 13° relative to the swimming direction. Thick vertical lines on (d) mark the extent of the cephalofoil and pectoral fins along the body. 'cb' marks the centre of buoyancy. $\bar{x} = x/l$ is the reduced coordinate along the body.

with increasing angle of attack. When both the cephalofoil and the pectoral fins stall, the centre of pressure becomes practically independent of their orientation relative to the body (figures 3d and 5d). With the cephalofoil removed, the centre of pressure stays around the aerodynamic centre of the pectoral fins (figure 4d).

Increasing the angle of the pectoral fins $\delta_{pf}$ increases the lift and the pitching moment (figure 5a,c) but hardly changes the centre of pressure at high angles of attack, when most of the lift is associated with the fins (figure 5d). The effect of $\delta_{pf}$ on the centre of pressure is opposite to that of $\delta_c$. Increasing $\delta_{pf}$ pulls the centre of pressure posteriorly, toward the fins; increasing $\delta_c$ pulls the centre of pressure anteriorly, toward the cephalofoil. The pectoral fins stall when their angle relative to the flow $\alpha + \delta_{pf}$ exceeds 13°. The (slightly) higher stall angle as compared with that of the cephalofoil can be attributed to several factors: unhedral angle, crescent planform, etc. For example, replacing the crescent fins with equivalent rectangular fins (having the same span, area and profile, electronic supplementary material, figure S2), reduced the stall angle by approximately 3.5° (figure 4a–c).

## 3.2. Minimal speed and centre-of-mass limits with $\lambda_{cf} = 0$

The scaled swim speed $\widehat{Fr}$ and the scaled distance between the centres of mass and buoyancy $\widehat{X}$ that balance the shark at constant depth are shown in figure 6a,b for a few combinations of $\delta_c$ and $\alpha$ when $\delta_{pf} = \lambda_{cf} = 0$, and in figure 7a,b for a few combinations of $\delta_{pf}$ and $\alpha$ when $\delta_c = \lambda_{cf} = 0$; additional cases can be found in the electronic supplementary material, figure S5a,b. Because the swim speed decreases with increasing lift coefficient (A 18), it generally decreases with increasing $\delta_c$, $\delta_{pf}$ or $\alpha$ (figures 6a and 7a; electronic supplementary material, figure S5a). The minimal scaled swim speed is, approximately, 0.7, and it is

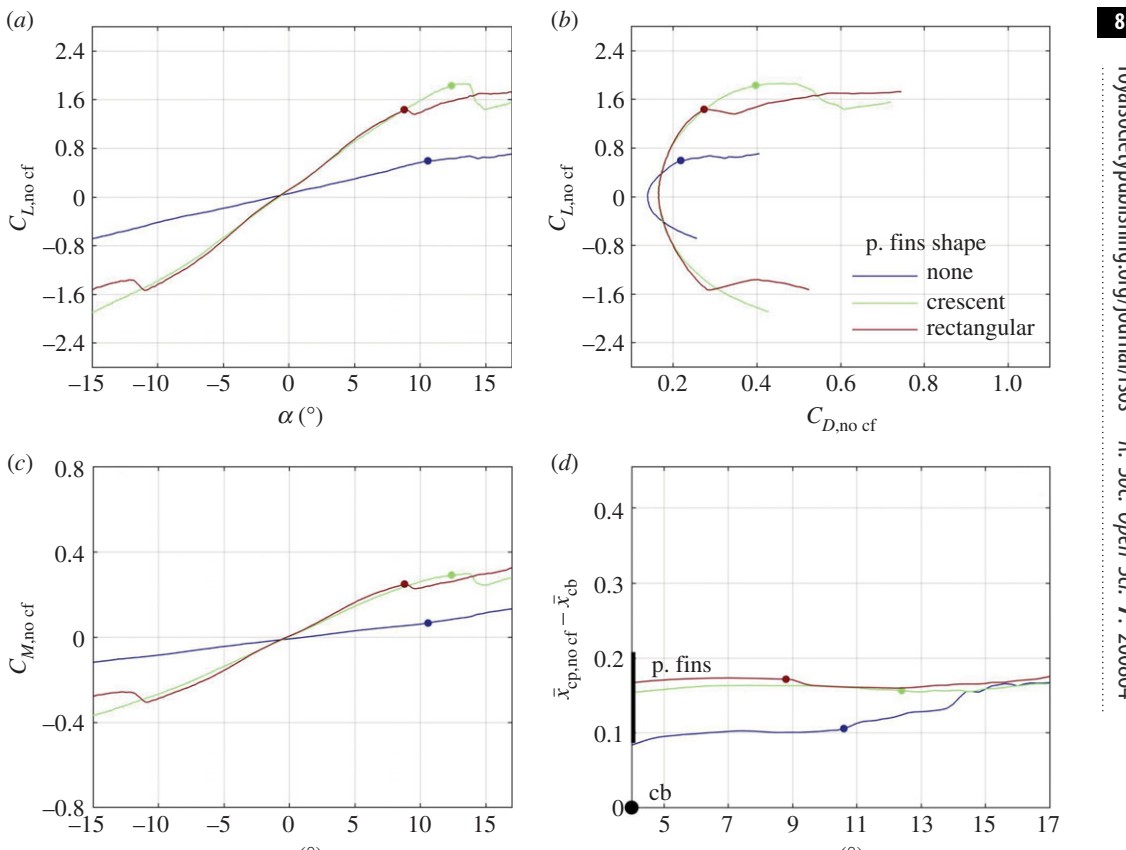

**Figure 4.** The same as figure 3 only now the cephalofoil has been removed by replacing the head with an ordinary looking one (electronic supplementary material, figures S1 and S2). Three configurations of pectoral fins were tested. The first one had the same (crescent) fins as in the experiments shown on figures 3 and 5, set at $\delta_{pf} = 0$. The second configuration had the crescent fins replaced by rectangular fins of the same area, span, and profile, also set at $\delta_{pf} = 0$. The third configuration had the pectoral fins completely removed. Its stall is associated with the stall of the pelvic fins (as manifested in an increase in the pitching moment, a decrease in lift, and a forward shift in the centre of pressure).

invariably associated with $\alpha \approx 13°$, the angle at which the pectoral fins, when set parallel to the caudo-cranial axis ($\delta_{pf} = 0$), stall (figures 6a and 7a; electronic supplementary material, figure S5a). For a 2.5 m shark with $\beta = 0.04$, the scaled speed numerically equals the swim speed in $m\,s^{-1}$ by (2.3).

As long as the pitching moment about the centre of buoyancy is positive (figures 3c and 5c; electronic supplementary material, figure S4c), the balancing centre-of-mass position is found anterior to the centre of buoyancy (figures 6b and 7b; electronic supplementary material, figure S5b). In general, increasing $\delta_c$ moves the balancing centre of mass anteriorly (figure 6b); increasing $\delta_{pf}$ moves it posteriorly (figure 7b) – both are consistent with the respective shift in the centre-of-pressure position (figures 3d and 5d). Stall of the pectoral fins at high angles of attack (this is the phenomenon seen in figures 6b and 7b) moves it anteriorly. Increasing the angle of attack reduces the possible range of the balancing centre-of-mass positions (figures 6b and 7b; electronic supplementary material, figure S5b).

The range of centre-of-mass positions that allows the shark to swim at any speed above the minimal speed extends between the most posterior (with respect to speed) of the most anterior (with respect to alignment angles of the control surfaces) balancing centre-of-mass positions and the most anterior (with respect to speed) of the most posterior (with respect to alignment angles of the control surfaces) balancing centre-of-mass positions (appendix A). It is practically a single point (0.25, in scaled units) for $\widehat{Fr} > 0.7$, but extends to the interval (0.21,0.29) for $\widehat{Fr} > 0.8$. This range of centre-of-mass positions applies when the cephalofoil is used as the primary control surface, assisted or not by the pectoral fins (figure 7b; electronic supplementary material, figure S5b). It reduces to (0.25, 0.29) when the pectoral fins are used as the primary surfaces, unassisted by the cephalofoil (figure 6b). For a 2.5 m shark with $\beta = 0.04$, $\widehat{X}$ numerically equals the physical distance in decimetres, so the viable range of centre-of-mass positions extends between 21 and 29 mm anterior to the centre of buoyancy—a diminutive 8 mm (3‰ body length) margin.

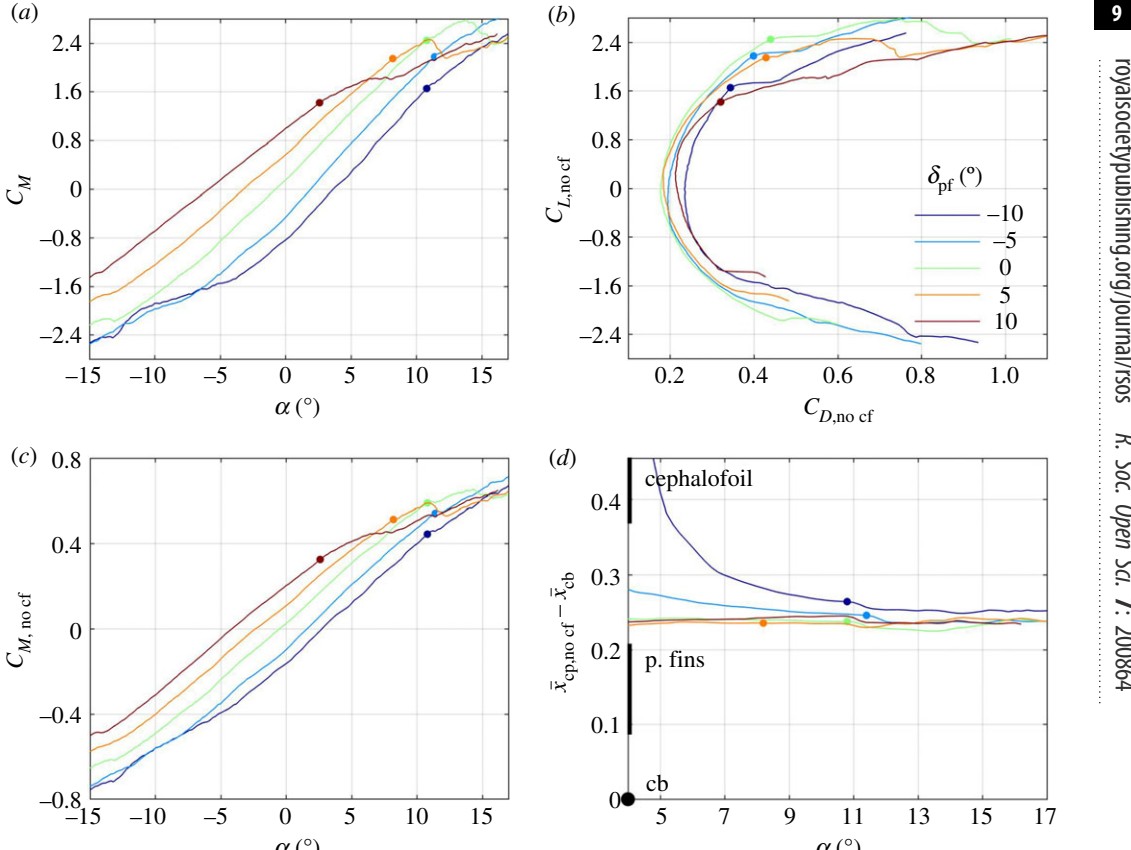

**Figure 5.** Same as figure 3 only now the cephalofoil is the one that is aligned with the caudo-cranial axis ($\delta_c = 0$), and the pectoral fins move between $-10°$ and $+10°$ relative to the body. The mid-line ($\delta_c = \delta_{pf} = 0$) is the same as the mid-line in figure 3.

### 3.3. Minimal speed and centre-of-mass limits with $\lambda_{cf} = 1$

An increase in $\lambda_{cf}$ reduces the minimal swim speed (compare figures 6a and 6c; 7a and 7c; electronic supplementary material, figures S5a and S5c), and shifts the balancing centre of mass posteriorly (compare figures 6b and 6d; 7b and 7d; electronic supplementary material, figure S5b and S5d). The magnitude of the effect is variable. When changing $\lambda_{cf}$ from 0 to 1, the minimal (scaled) swim speed decreases by approximately 0.1; the (scaled) balancing centre-of-mass position shifts posteriorly by approximately 0.2 at the minimal speed, but many times more than that at high speed. In fact, there is no viable range of centre-of-mass positions that allow balancing the shark with $\lambda_{cf} = 1$ and any $\widehat{Fr} > 0.8$: if balanced at low speeds, the shark tumbles at high speeds.

### 3.4. Minimal speed and centre-of-mass limits during descent

With increasing angle of descent (negative $\gamma$), the thrust needed to sustain speed of a negatively buoyant shark gradually diminishes, and vanishes at $\gamma = -\tan^{-1}(C_{D,\text{no cf}}/C_{L,\text{no cf}})$. Because the drag-to-lift ratio of the shark is a few tenths (figures 3b, 5b), this free-glide dive angle is sufficiently small to render $\cos\gamma \approx 1$. Adopting the assumption that the lift of the caudal fin is proportional to its thrust, vanishing thrust implies vanishing lift. Consequently, the minimal speed and the centre-of-mass limits in an unpowered dive are practically the same as those when swimming at constant depth with $\lambda_{cf} = 0$— compare electronic supplementary material, figures S5a,b and S6a,b.

### 3.5. Minimal speed and centre-of-mass limits during ascent

With increasing angle of ascent (positive $\gamma$), the thrust needed to sustain speed gradually increases, and, in principle, the shark can hover at zero swim speed. An example can be found in electronic supplementary material, figure S7. At low speeds, however, hydrodynamic forces that can be generated by the cephalofoil and the pectoral fins are small, so the shark must rely solely on the caudal fin for longitudinal control.

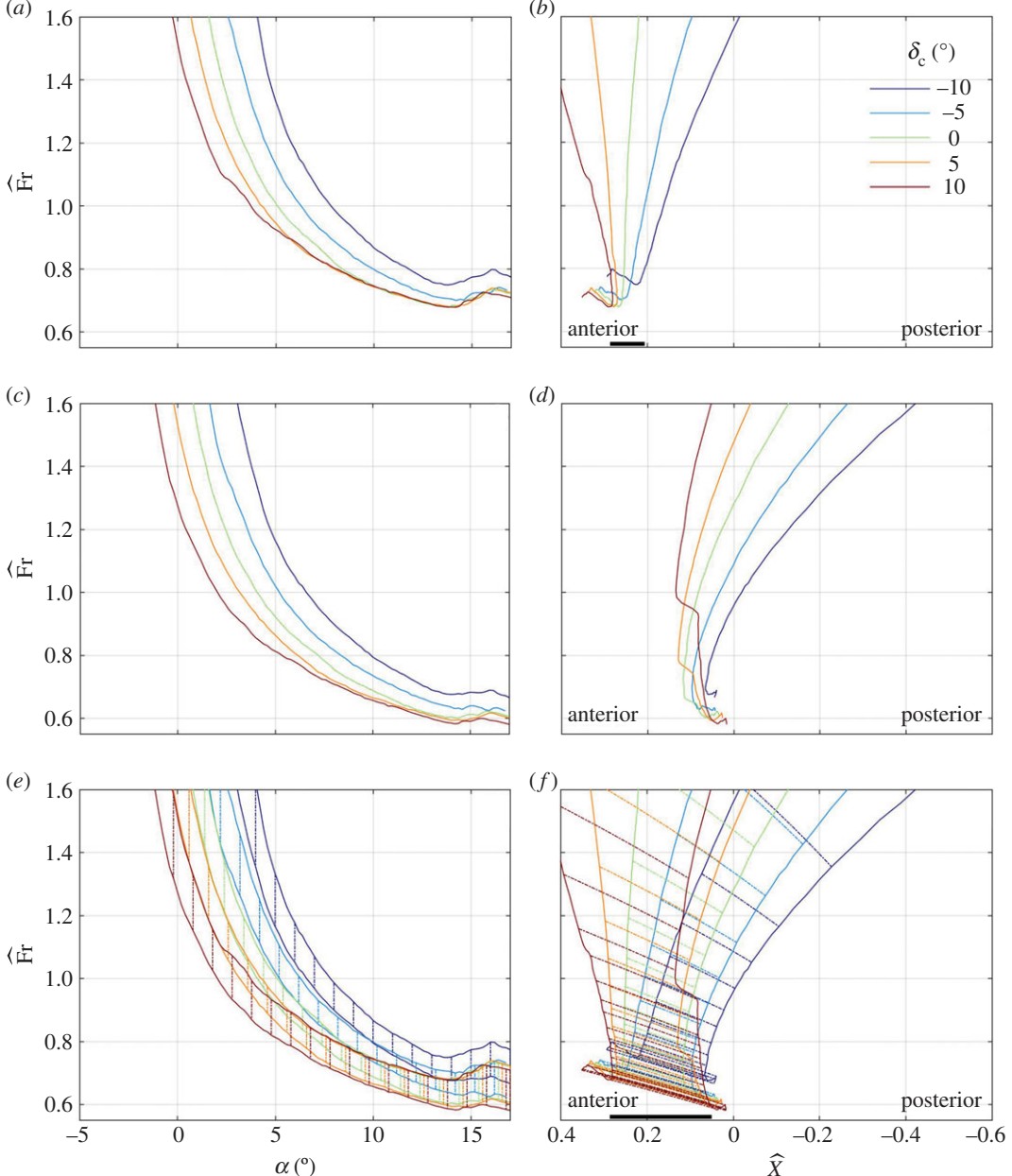

**Figure 6.** The scaled speed ($a,c,e$) and the scaled centre-of-mass margin ($b,d,f$)—positive when the centre of mass is anterior to the centre of buoyancy—that balance a shark at $\gamma = 0$ with $\lambda_{cf} = 0$ ($a,b$), $\lambda_{cf} = 1$ ($c,d$), $\lambda_{cf} \in (0, 1)$ ($e,f$) for different alignment angles of the cephalofoil ($\delta_c$) when $\delta_{pf} = 0$. Vertical lines on ($e$) and slanted (dash-dotted) lines on ($f$) connect the balance points with $\lambda_{cf} = 0$ to the respective balance points with $\lambda_{cf} = 1$. Thick horizontal lines on ($b$) and ($f$) mark the range of centre-of-mass positions that allow swimming at any $\widehat{Fr} > 0.8$; this range does not exist when $\lambda_{cf} = 1$. For a 2.5 m shark with $\beta = 0.04$, $\widehat{Fr}$ is numerically the same as the swim speed in m s$^{-1}$; $\widehat{X}$ is numerically the same as the distance between the centres of mass and buoyancy in decimetres. This figure is based on the data shown in figure 3.

# 4. Discussion

For the sake of argument, let us return to the case where the shark swims at constant depth ($\gamma = 0$), generating no lift with its caudal fin ($\lambda_{cf} = 0$). Introducing $\gamma = \lambda_{cf} = 0$ in (2.3) and (2.4) yields

$$\widehat{Fr} = \sqrt{\frac{2k_{pc}}{C_{L,\text{no cf}}}} \tag{4.1}$$

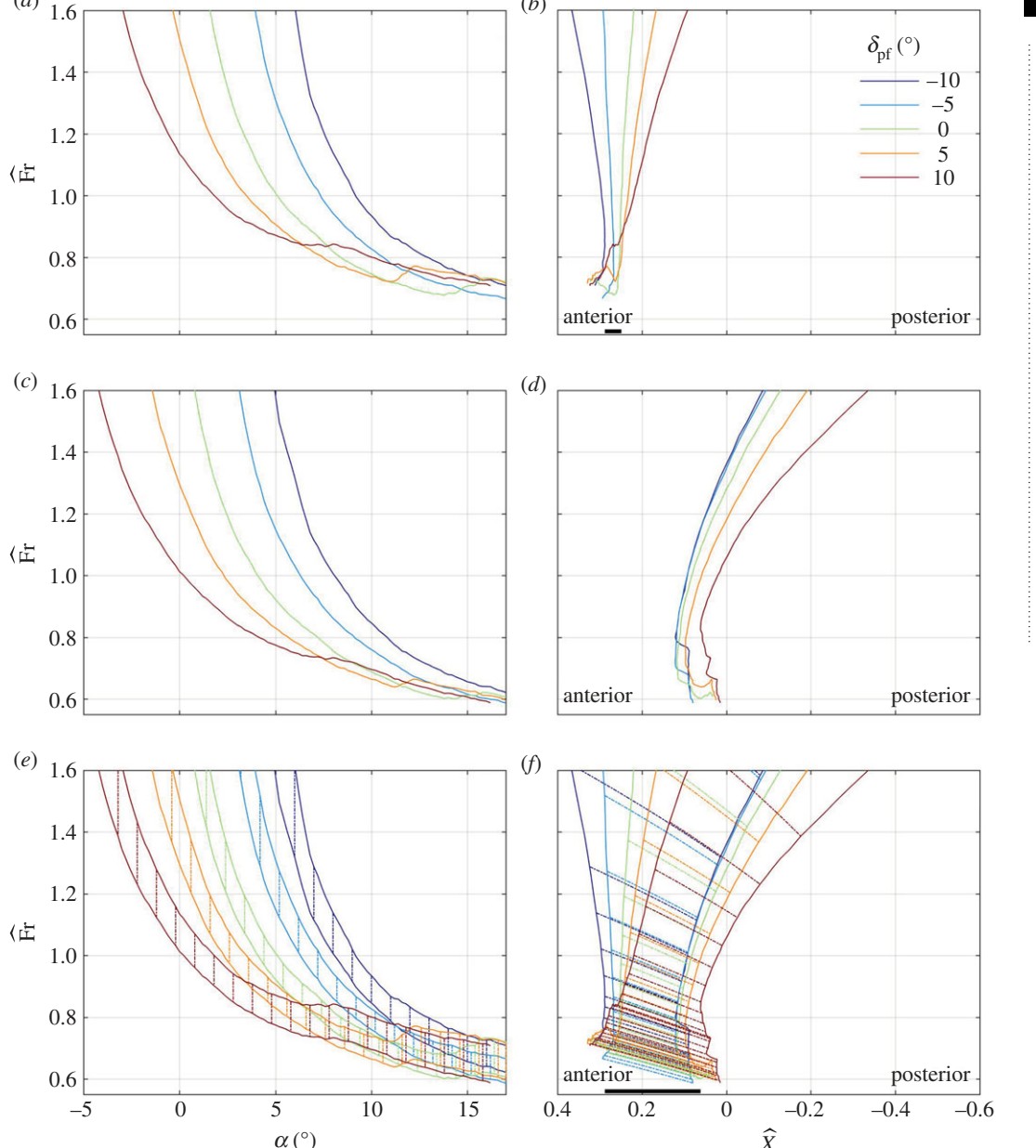

**Figure 7.** (a–f) Same as figure 6, only the pectoral fins move rather than the cephalofoil, which is held here at $\delta_c = 0$. This figure is based on the data shown in figure 5.

for the scaled balancing speed, and

$$\widehat{X} = (\bar{x}_{cp,no\,cf} - \bar{x}_{cb})\cos\alpha + \frac{C_{D,no\,cf}}{C_{L,no\,cf}}\left((\bar{x}_{cp,no\,cf} - \bar{x}_{cf})\sin\alpha - \cos\alpha(\bar{z}_{cb} - \bar{z}_{cf})\right) \qquad (4.2)$$

for the scaled balancing margin between the centres of mass and buoyancy.

Anterior and posterior limits of the centre of mass are invariably associated with swimming at the minimal speed, on the verge of stall of the pectoral fins (figures 6b and 7b). At these conditions, both the angle of attack (in radians) and the drag-to-lift ratio are a few tenths (figures 3a,b and 5a,b), and, because $\bar{z}_{cb} - \bar{z}_{cf}$ is a few hundredths (figure 1), $\widehat{X}$ can be approximated by

$$\widehat{X} \approx \bar{x}_{cp,no\,cf} - \bar{x}_{cb}. \qquad (4.3)$$

In other words, the minimal swim speed is determined by the maximal lift coefficient at which the pectoral fins stall, whereas the anterior and posterior limits of the centre of mass are determined by the extremal positions of the centre of pressure when the pectoral fins are on the verge of stall.

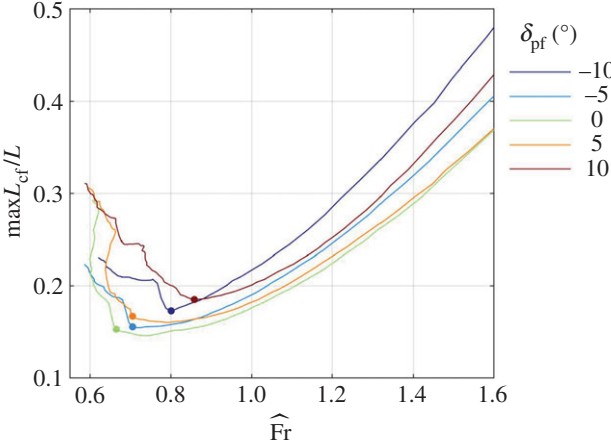

**Figure 8.** The share of the caudal fin in generation of lift when $\lambda_{cf} = 1$ and $\gamma = 0$. Based on the data shown in figure 5 ($\delta_c = 0$). The centre of mass is not fixed and moves to balance the shark for each combination of $\delta_{pf}$ and $\widehat{Fr} = v/\sqrt{g\beta l}$. The great hammerhead cruises at $\widehat{Fr} \approx 0.8$ [9].

When swimming on the verge of stall of the pectoral fins, there is not much movement of the centre of pressure with the change of orientation of the pectoral fins relative to the body of the shark (figure 5d), and hence the use of pectoral fins (and pectoral fins only) for control results in practically zero margin between anterior and posterior centre-of-mass limits (figure 7b). Extension of the neck (that increases $\delta_c$) is also ineffective (figure 3d), because the cephalofoil stalls even before the pectoral fins do (figure 3a,b). On the other hand, flexion of the neck (that reduces $\delta_c$) turns out to be the most effective control at high angles of attack by moving the centre of pressure posteriorly (figure 3d). Flexion of 10° shifts it by 0.04 (figure 3d) and one could expect that additional 10° will double the shift. Flexion beyond 22°–23°, however, loses the effect by causing a ventral-side stall of the cephalofoil (similar to its stall when set at $\delta_c = -10°$ and $\alpha \approx -2.5°$; although left unmarked, this point is clearly identifiable on figure 3a,b). Apart from generating pitching down moment, flexion of the neck also reduces the maximal lift coefficient: 10° flexion reduces it by almost 0.5, a mere 18% (figure 3a), which is equivalent to an increase of almost 10% in the minimal swim speed (equation (4.1), figure 6a). Doubling the flexion will double the effect, and hence balancing the shark with cephalofoil when its centre of mass is at the posterior limit has a large performance penalty. In any case, the minimal (scaled) swim speed of 0.7 and the (scaled) margin of 0.08 between the anterior and posterior centre-of-mass limits for swimming at scaled speeds above 0.8, that were cited in §3.2, are backed up by additional combinations of $\delta_c$, $\delta_{pf}$ and $\alpha$ found in the electronic supplementary material, figure S5a,b. This is 3‰ body length when $\beta = 0.04$.

Minimal swim speed observed with a free swimming 2.3 m shark (2.95 m TL) was 0.6 m s$^{-1}$ (see supplementary material to [9]). Assuming $\beta$ between 0.04 and 0.05 [15], it implies that this shark swam at scaled speed between 0.56 and 0.63. It is shown in appendix B that if it could fill the mouth with a large chunk of fat, its centre of mass would move approximately the same 3‰ body length as the margin between the anterior and posterior centre-of-mass limits. Numbers simply do not fit.

Letting the caudal fin generate lift in a constant proportion to thrust, rectifies the minimal swim speed discrepancy (figures 6c and 7c), but makes the shark tumble at high speeds (figures 6d and 7d). A naive explanation can be based on the notion that drag of the shark increases with speed, while its hydrodynamic lift remains equal to the excess weight and hence remains constant. Because thrust offsets drag, and lift of the caudal fin is proportional to thrust, the lift of the caudal fin increases with speed as well. An increasing share of the caudal fin in generation of lift (equation (A 12); figure 8) implies a diminishing share of the cephalofoil and the pectoral fins, and the pitching moment generated by it overtakes the counteracting moment of the cephalofoil and the pectoral fins.

Letting the caudal fin generate lift with variable lift-to-thrust ratio, rectifies all issues, both by reducing the minimal swim speed, and by extending the viable range of the centre-of-mass positions (figures 6e,f and 7e,f; electronic supplementary material, figure S5e,f). With $\lambda_{cf}$ taking on any value in the interval (0,1), the viable range of centre-of-mass range becomes (–0.04, 0.29). This is only a 30 mm margin for a 2.5 m shark with $\beta = 0.04$ (1.3% body length), but not inconceivable. In other words, a shark must have a way to control the lift-to-thrust ratio of its caudal fin. A possible way to change $\lambda_{cf}$ is by varying the twist of the dorsal lobe, as a sea snake does [16], or by changing the lateral flex of the ventral lobe [3].

But what happens in an unpowered glide? With any finite $\lambda_{cf}$, zero thrust implies zero lift, and when the glide angle is sufficiently small, this case becomes equivalent to the case $\gamma = \lambda_{cf} = 0$, just rendered unlikely. Either a great hammerhead cannot glide idle, or, what is much more probable, it can generate lift by 'dragging' the tail—that is, it can generate $\lambda_{cf} < 0$. It is noted that if the shark can also generate sufficiently large negative $\lambda_{cf}$ when swimming actively, it can practically double the viable range of centre-of-mass positions and extend it anteriorly.

# 5. Concluding remarks

Using a variant of *reductio ad absurdum*, we have shown that the caudal fin of the great hammerhead (and, by extension, any shark) should generate some lift to balance it. This result accords with previous studies that were based on analyses of posture and wake structure of actively swimming leopard [2–4] and bamboo [2] sharks. We have also shown that the lift of the caudal fin cannot be proportional to its thrust, and the ratio of the two must change with the swim speed. Changing lift-to-thrust ratio implies that a shark has a way to actively control the deformation (flex or twist) of the caudal fin. The share of the caudal fin in the total hydrodynamic lift generated by the shark varies with location of the centre of mass, trajectory angle relative to horizon, and alignment angles of the pectoral fins and the cephalofoil relative to the body, but, in general, cannot be significantly larger than the ratio of thrust and submerged weight of the shark. It is 15–20% at a typical cruise speed.

Perhaps the most conspicuous result of this study is the diminutive range of viable centre-of-mass locations. Considering inhomogeneity of tissues comprising the shark's body [17] and possible (seasonal and ontogenetic) variations in size and density of its liver, it seems rather unlikely that the location of the centre of mass can fall within a few tens of millimetres of the centre of volume by chance. The same applies to other pelagic species lacking a swim bladder, ocean sunfish (*Mola mola*) in particular [18]. Its ability to lay practically motionless just below the water surface [19] implies that its centres of mass and buoyancy practically coincide.

Data accessibility. All data underlying this study can be found on the Dryad Digital Repository at https://doi.org/10.5061/dryad.xwdbrv1b8 [20]. It includes both the experimental data and printer-ready files of the wind-tunnel model.
Competing interests. I declare I have no competing interests
Funding. I received no funding for this study
Acknowledgements. The author thanks the staff of the Aerodynamics Laboratory at the Faculty of Aerospace Engineering, Technion, for their assistance with the wind-tunnel experiments.

# Appendix A. Longitudinal balance of a shark

## A.1. Assumptions

The central assumption underlying derivations of this appendix is that the shark swims steadily in an upright posture along a straight path, and hence the totality of forces and moments acting on it vanish. In so far as the hydrodynamic forces are concerned, it is assumed that the caudal fin generates mainly thrust and lift, whereas the rest of the shark generates mainly lift and drag (these are tail-beat-averaged forces). In other words,

$$T = T_{cf}, \quad D = D_{no\,cf}, \tag{A1}$$

but

$$L = L_{no\,cf} + L_{cf}, \quad M_y = M_{y,no\,cf} + M_{y,cf}, \tag{A2}$$

where the subscripts 'cf' and 'no cf' mark the respective contributions. It is also assumed that the force generated by the caudal fin can be associated with a single point $(x_{cf}, 0, z_{cf})$ in body-fixed reference frame $R^B$ (figure 1). Consequently, the pitching moment it generates about the centre of buoyancy $(x_{cb}, 0, z_{cb})$ is

$$M_{y,cf} = -(L_{cf}\cos\alpha - T_{cf}\sin\alpha)(x_{cb} - x_{cf}) - (L_{cf}\sin\alpha + T_{cf}\cos\alpha)(z_{cb} - z_{cf}), \tag{A3}$$

where $\alpha$ is the angle of attack. The ratio of lift and thrust of the caudal fin,

$$\lambda_{cf} = \frac{L_{cf}}{T_{cf}}, \tag{A4}$$

is, essentially, the definition of $\lambda_{cf}$.

## A.2. Balance requirements

To balance a shark swimming in an upright posture along a straight path, the (tail-beat-averaged) hydrodynamic and hydrostatic forces acting on it in the $x$-$z$ plane should cancel out with gravity, whereas the (tail-beat-averaged) hydrodynamic pitching moment should cancel out with gravity–buoyancy couple. Using (A 1) and (A 2), these requirements can be formally written as

$$T_{cf} = D_{no\,cf} + \beta G \sin \gamma, \tag{A 5}$$

$$L = \beta G \cos \gamma \tag{A 6}$$

and

$$M_y = G(x^E_{cm} - x^E_{cb}), \tag{A 7}$$

where

$$\beta = \frac{G - B}{G} \tag{A 8}$$

is the ratio between weights of the shark in and out of water, whereas

$$x^E_{cm} - x^E_{cb} = (x_{cm} - x_{cb}) \cos (\alpha + \gamma) + (z_{cm} - z_{cb}) \sin (\alpha + \gamma) \tag{A 9}$$

is the horizontal projection of the distance between centres of mass and buoyancy. Equation (A 5) manifests the balance of forces along the $x$-axis of $R^S$; equation (A 6) manifests the balance of forces along the $z$-axis of $R^S$, and equation (A 7) manifests the balance of the pitching moment about the centre of buoyancy. When swimming upright, the three additional balance requirements (the sum of forces along the $y$-axis and the sum of moments about the $x$- and $z$-axes) become redundant.

## A.3. Balancing swim speed and mass–buoyancy margin

The three balance requirements (A 5)–(A 7) relate eight parameters: swim-path angle, angle of attack, alignment angles of the cephalofoil and the pectoral fins, lift to thrust ratio of the caudal fin, swim speed, thrust and the horizontal projection of the margin between the centres of mass and buoyancy, and hence five of them are free. In what follows, the free parameters will be the first five, and (A 5)–(A 7) will be satisfied by adjusting swim speed, thrust and the distance between centres of mass and buoyancy. It loosely resembles balancing a hang-glider.

The forces and moments acting on the body of the shark (they are marked by the subscript 'no cf') can be reconstructed directly from wind-tunnel experiments as products of the dynamic pressure $q = (1/2)\rho v^2$ ($\rho$ here stands for density of water and $v$ for the swim speed) and certain dimensionless coefficients depending on the angle of attack, and cephalofoil and pectoral fins alignment angles relative to the body (2.1). The forces generated by the caudal fin (they are marked by the subscript 'cf') are a part of the solution—thrust follows by (A 5), whereas the lift and pitching moment follow thrust by (A 4) and (A 3).

Forces acting on the entire shark are

$$L = L_{cf} + L_{no\,cf} = \cos \gamma \frac{L_{no\,cf} + \lambda_{cf} D_{no\,cf}}{\cos \gamma - \lambda_{cf} \sin \gamma} \tag{A 10}$$

and

$$M_y = M_{y,no\,cf} + M_{y,cf} = \frac{L}{\cos \gamma} \left\{ \frac{M_{y,no\,cf}(\cos \gamma - \lambda_{cf} \sin \gamma)}{L_{no\,cf} + \lambda_{cf} D_{no\,cf}} \right.$$
$$\left. - \frac{L_{no\,cf} \sin \gamma + D_{no\,cf} \cos \gamma}{L_{no\,cf} + \lambda_{cf} D_{no\,cf}} ((\lambda_{cf} \cos \alpha - \sin \alpha)(x_{cb} - x_{cf}) + (\lambda_{cf} \sin \alpha + \cos \alpha)(z_{cb} - z_{cf})) \right\}. \tag{A 11}$$

Equation (A 10) follows (A 2) using

$$L_{cf} = \beta G \cos \gamma - L_{no\,cf} = \lambda_{cf} \frac{L_{no\,cf} \sin \gamma + D_{no\,cf} \cos \gamma}{\cos \gamma - \lambda_{cf} \sin \gamma}; \tag{A 12}$$

equation (A 11) follows (A 2) using (A 10) and (A 12). In turn, equation (A 12) can be obtained from (A 6) with substitution of (A 2), followed by substitution of $\beta W$ from (A 5), and of $T_{cf}$ from (A 4).

Equation (A 12) yields the balancing thrust $T = T_{cf} = L_{cf}/\lambda_{cf}$ by (A 1) and (A 4). The balancing speed and the distance between centres of mass and buoyancy follow (A 6) and (A 7) by (A 10), (A 11) and

(2.1). Equation (A 6) yields the speed

$$v = \widehat{\mathrm{Fr}}\sqrt{\frac{g\beta l}{1-\beta}} \approx \widehat{\mathrm{Fr}}\sqrt{g\beta l};$$ (A 13)

equation (A 7) yields the distance between centres of mass and buoyancy,

$$x_{\mathrm{cm}}^E - x_{\mathrm{cb}}^E = \beta l \widehat{X}.$$ (A 14)

In these, $g$ is the acceleration of gravity, whereas

$$\widehat{\mathrm{Fr}} = \sqrt{2k_{\mathrm{pc}}\frac{\cos\gamma - \lambda_{\mathrm{cf}}\sin\gamma}{C_{L,\mathrm{no\,cf}} + \lambda_{\mathrm{cf}}C_{D,\mathrm{no\,cf}}}}$$ (A 15)

and

$$\widehat{X} = \frac{C_{M,\mathrm{no\,cf}}(\cos\gamma - \lambda_{\mathrm{cf}}\sin\gamma)}{C_{L,\mathrm{no\,cf}} + \lambda_{\mathrm{cf}}C_{D,\mathrm{no\,cf}}} - \frac{C_{D,\mathrm{no\,cf}}\cos\gamma + C_{L,\mathrm{no\,cf}}\sin\gamma}{C_{L,\mathrm{no\,cf}} + \lambda_{\mathrm{cf}}C_{D,\mathrm{no\,cf}}}$$
$$\times ((\lambda_{\mathrm{cf}}\cos\alpha - \sin\alpha)(\bar{x}_{\mathrm{cb}} - \bar{x}_{\mathrm{cf}}) + (\lambda_{\mathrm{cf}}\sin\alpha + \cos\alpha)(\bar{z}_{\mathrm{cb}} - \bar{z}_{\mathrm{cf}}))$$ (A 16)

are the respective scaled quantities. The neglect of $\beta$ in the denominator of (A 13) is justified because its typical value is a few hundredths [15]. In (A15),

$$k_{\mathrm{pc}} = B/(\rho g S l)$$ (A 17)

is the prismatic coefficient, the ratio between the volume of the shark and the minimal cylinder enclosing its body (with no fins); based on the CAD model, $k_{\mathrm{pc}} \approx 0.64$. In (A 16), $\bar{x} = x/l$ and $\bar{z} = z/l$ are the respective reduced coordinates.

## A.4. Variants

In spite their relative complexity, equations (A 15) and (A 16) have the advantage of furnishing the scaled swim speed and the scaled margin between centres of mass and buoyancy directly from the wind-tunnel data. Their alternative forms,

$$\widehat{\mathrm{Fr}} = \sqrt{\frac{2k_{\mathrm{pc}}\cos\gamma}{C_L}}$$ (A 18)

and

$$\widehat{X} = (\cos\gamma/C_L)\Big(C_{M,\mathrm{no\,cf}} - C_{L,\mathrm{cf}}\big(\cos\alpha(\bar{x}_{\mathrm{cb}} - \bar{x}_{\mathrm{cf}}) + \sin\alpha(\bar{z}_{\mathrm{cb}} - \bar{z}_{\mathrm{cf}})\big)$$
$$- C_{T,\mathrm{cf}}\big(-\sin\alpha(\bar{x}_{\mathrm{cb}} - \bar{x}_{\mathrm{cf}}) + \cos\alpha(\bar{z}_{\mathrm{cb}} - \bar{z}_{\mathrm{cf}})\big)\Big),$$ (A 19)

which immediately follow from (A 6) and (A 7) using (A 2), (A 3) and (2.1), are much simpler, but they are not practical, because they involve unknown contributions of the caudal fin.

## A.5. Finding the centre-of-mass and minimal speed limits

Wind-tunnel experiments furnish $C_{L,\mathrm{no\;cf}}$, $C_{D,\mathrm{no\;cf}}$ and $C_{M,\mathrm{no\;cf}}$ as functions of $\alpha$, $\delta_{\mathrm{pf}}$ and $\delta_{\mathrm{c}}$. Given $\lambda_{\mathrm{cf}}$ and $\gamma$, equations (A 15) and (A 16) furnish the scaled swim speed $\widehat{\mathrm{Fr}}(\alpha,\delta_{\mathrm{c}},\delta_{\mathrm{pf}};\lambda_{\mathrm{cf}},\gamma)$ and the scaled margin between the centres of mass and buoyancy $\widehat{X}(\alpha,\delta_{\mathrm{c}},\delta_{\mathrm{pf}};\lambda_{\mathrm{cf}},\gamma)$ that balance the shark. The minimal scaled swim speed that can be obtained with all viable combinations of $\alpha$, $\delta_{\mathrm{pf}}$ and $\delta_{\mathrm{c}}$,

$$\widehat{\mathrm{Fr}}_{\min}(\lambda_{\mathrm{cf}},\gamma) = \min_{\alpha,\delta_{\mathrm{pf}},\delta_{\mathrm{c}}} \widehat{\mathrm{Fr}}(\alpha,\delta_{\mathrm{c}},\delta_{\mathrm{pf}};\lambda_{\mathrm{cf}},\gamma),$$ (A 20)

sets the minimal swim-speed limit $v_{\min}(\lambda_{\mathrm{c}},\gamma) \approx \widehat{\mathrm{Fr}}_{\min}(\lambda_{\mathrm{c}},\gamma)\sqrt{gl\beta}$ with those $\lambda_{\mathrm{cf}}$ and $\gamma$. Concurrently, the extremes,

$$\widehat{X}_{\mathrm{a}}(\widehat{\mathrm{Fr}}',\lambda_{\mathrm{cf}},\gamma) = \max_{\delta_{\mathrm{pf}},\delta_{\mathrm{c}},\widehat{\mathrm{Fr}}(\alpha,\delta_{\mathrm{c}},\delta_{\mathrm{pf}};\lambda_{\mathrm{cf}},\gamma)=\widehat{\mathrm{Fr}}'} \widehat{X}(\alpha,\delta_{\mathrm{c}},\delta_{\mathrm{pf}};\;\lambda_{\mathrm{cf}},\gamma)$$ (A 21)

and

$$\widehat{X}_{\mathrm{p}}(\widehat{\mathrm{Fr}}',\lambda_{\mathrm{cf}},\gamma) = \min_{\delta_{\mathrm{pf}},\delta_{\mathrm{c}},\widehat{\mathrm{Fr}}(\alpha,\delta_{\mathrm{c}},\delta_{\mathrm{pf}};\lambda_{\mathrm{cf}},\gamma)=\widehat{\mathrm{Fr}}'} \widehat{X}(\alpha,\delta_{\mathrm{c}},\delta_{\mathrm{pf}};\lambda_{\mathrm{cf}},\gamma), \tag{A 22}$$

set the respective anterior and posterior centre-of-mass limits, $x_{\mathrm{cb}}^E + \beta l \widehat{X}_{\mathrm{a}}(\widehat{\mathrm{Fr}}',\lambda_{\mathrm{cf}},\gamma)$ and $x_{\mathrm{cb}}^E + \beta l \widehat{X}_{\mathrm{p}}(\widehat{\mathrm{Fr}}',\lambda_{\mathrm{cf}},\gamma)$ at the given scaled swim speed $\widehat{\mathrm{Fr}}' > \widehat{\mathrm{Fr}}_{\min}(\lambda_{\mathrm{cf}},\gamma)$. To be able to swim at any $\widehat{\mathrm{Fr}}' > \widehat{\mathrm{Fr}}_{\min}(\lambda_{\mathrm{cf}},\gamma)$, the centre of mass should be located between the most posterior of the anterior limits, $x_{\mathrm{cb}}^E + \min_{\widehat{\mathrm{Fr}}'>\widehat{\mathrm{Fr}}_{\min}(\lambda_{\mathrm{cf}},\gamma)} \beta l \widehat{X}_{\mathrm{a}}(\widehat{\mathrm{Fr}}',\lambda_{\mathrm{cf}},\gamma)$, and the most anterior of the posterior limits, $x_{\mathrm{cb}}^E + \max_{\widehat{\mathrm{Fr}}'>\widehat{\mathrm{Fr}}_{\min}(\lambda_{\mathrm{cf}},\gamma)} \beta l \widehat{X}_{\mathrm{p}}(\widehat{\mathrm{Fr}}',\lambda_{\mathrm{cf}},\gamma)$. This paradigm underlies the ranges shown in figures 6a–d and 7a–d, as well as electronic supplementary material, figures S3a–d, S4a,b and S5a–d. The ranges shown in figures 6e,f, 7e,f, and in electronic supplementary material, figure S5e,f, extend between the anterior limit for $\lambda_{\mathrm{cf}} = 0$ and the posterior limit with $\lambda_{\mathrm{cf}} = 1$.

# Appendix B. Centre of mass

The centre of mass of a shark can be formally defined by

$$x_{\mathrm{cm}} = \frac{1}{m} \int_\Omega \rho_{\mathrm{b}}(\mathbf{r}) x \, \mathrm{d}^3\mathbf{r}, \tag{B 1}$$

where $\rho_{\mathrm{b}}$ is the density of its body, $\Omega$ is the region in space occupied by it, and

$$m = \int_\Omega \rho_{\mathrm{b}}(\mathbf{r}) \mathrm{d}^3\mathbf{r} \tag{B 2}$$

is the respective mass. A variation $\delta\rho_{\mathrm{b}}$ in the body density yields the variation

$$\delta m = \int_\Omega \delta\rho_{\mathrm{b}}(\mathbf{r}) \, \mathrm{d}^3\mathbf{r} \tag{B 3}$$

in the body mass, and the variation

$$\delta x_{\mathrm{cm}} = \frac{1}{m} \int_\Omega \delta\rho_{\mathrm{b}}(\mathbf{r}) x \, \mathrm{d}^3\mathbf{r} - \frac{1}{m^2} \int_\Omega \rho_{\mathrm{b}}(\mathbf{r}) x \, \mathrm{d}^3\mathbf{r} \int_\Omega \delta\rho_{\mathrm{b}}(\mathbf{r}) \mathrm{d}^3\mathbf{r} \tag{B 4}$$

in its centre-of-mass position. The former follows (B 2); the latter follows (B 1) and (B 2). Noting (B 1), equation (B 4) can be recast as

$$\delta x_{\mathrm{cm}} = \frac{1}{m} \int_\Omega \delta\rho_{\mathrm{b}}(\mathbf{r})(x - x_{\mathrm{cm}}) \, \mathrm{d}^3\mathbf{r}, \tag{B 5}$$

and, if the variation of density is localized at the distance $\Delta x$ from the centre of mass, it yields

$$\delta x_{\mathrm{cm}} \approx \frac{\delta m}{m} \Delta x. \tag{B 6}$$

Replacing water, normally filling the mouth of the shark (say, 10% of the body volume, one-third fork-lengths anterior to the centre of mass), by the same volume of fat (with, say, 10% lower density), should move the centre of mass posteriorly by approximately 3‰ fork-lengths.

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
