## [Reviewer comments · Royal Society Open Science]

Review History

RSOS-200864.R0 (Original submission)

Review form: Reviewer 1

Is the manuscript scientifically sound in its present form?

Yes

Are the interpretations and conclusions justified by the results?

Yes

Is the language acceptable?

Yes

Do you have any ethical concerns with this paper?

No

Have you any concerns about statistical analyses in this paper?

No

Recommendation?

Accept with minor revision (please list in comments)

Comments to the Author(s)

See the attached document (Appendix A).

Review form: Reviewer 2**Is the manuscript scientifically sound in its present form?**

Yes

Are the interpretations and conclusions justified by the results?

Yes

Is the language acceptable?

Yes

Do you have any ethical concerns with this paper?

No

Have you any concerns about statistical analyses in this paper?

Yes

Recommendation?

Major revision is needed (please make suggestions in comments)

Comments to the Author(s)

Summary:

The author performs wind-tunnel testing of a 3D printed model of a hammerhead shark with varying body angle, cephalofoil angle, and pectoral fin angles in order to determine the minimal swimming speed, as well as the anterior and posterior center-of-mass limits that allow the shark to remain longitudinally balanced across the whole range of possible swimming speeds.

While there have been several studies on hydrodynamics of shark swimming, we still do not know how the physical design of specific shark (such as the hammerhead) can provide better maneuvering capability. Studies using modeling approaches as in this manuscript can help us to identify importance of specific morphological variables.

Before I can recommend the manuscript for publication, revisions are needed to improve readability in both the text and in the figures. Also, clarifications on some of the statements/assumptions are needed and the discussion section needs to be significantly improved. Specific comments are provided below.

Abstract

- 'maximal margin between anterior and posterior center-of-mass limits turned out to be a diminutive $0.3(\beta)l$.

How diminutive is this? What length, or percentage of body length, is $0.3(\beta)l$ for a typical adult?

Introduction

- It would be helpful to provide an overview to the broad problem via discussion of previous studies. Background citations are strewn in places but there is no cohesive discussion of what we know and what remains unclear. Finally, the author uses the introduction to provide backing to more of the finer technical aspects that are introduced later in the study, making it difficult to read the manuscript from start to finish.

- Figure 1: Center of buoyancy location is given, and the distance between center-of-mass and center of buoyancy is stated to be exaggerated more than 10x.

Is this based on your limits for center-of-mass or based on previous work? If based on previous work, please indicate the location.

If L and D are supposed to be defined relative to the body orientation, the labels are slightly off (e.g., D_no cf is not in line with cranio-caudal axis).

Finally, please define λ_{cf} in the caption.

- The use of gravity as a force, and weight as the difference between gravity and buoyancy can make the manuscript hard to follow at times.
- "There is no difference between swimming in water and flying in air as long as the respective Reynolds numbers are similar and no cavitation occurs."

If the Mach number approaches the transonic region, but the Reynolds number is similar then there is a difference. This is an extreme case, but so is cavitation.

In addition, the Strouhal number is important in unsteady flows that are generated by fin motion. However, static wind tunnel models as in this study do not provide true similarity with dynamic organism-swimming. Please address this.

Materials and methods

- "all fins had NACA 0015 profile"
Why was this profile chosen? Is it similar to hammerhead shark fins (if so cite papers that support such a claim)? Easier to manufacture?

- "Both forces were assumed to act at $(x_{cf}, 0, z_{cf})$, 0.685l posterior to the center-of-buoyancy"

x, y, and z are not defined as a coordinate system. This could be added to figure 1.

- The definitions for the scaled swimming speed \hat{Fr} and scaled horizontal margin between the center of mass and center of buoyancy \hat{x} are not well explained in the main text of the paper, and rely heavily on the reader examining the appendix for explanation of their usage.

- Figure 2: please label components and show additional views for clarity (e.g., only 1 pectoral fin is shown in existing drawing).

Results

- Given that this was mainly an experimental study, it is important to include standard deviation or standard error in charts. It is possible to either show them in the form of error bars or shading of lines.
- "In general, both the lift and the pitching moment coefficients increase with the angle-of-attack and with the angle of the cephalofoil relative to the body, but the lift coefficient drops when the angle-of-attack exceeds 13 degrees"

Does this mean that the shark is dynamically unstable in the longitudinal direction, and has to constantly expend energy to keep swimming in a straight line instead of tumbling?

- "Consequently, all configurations with $\Delta_c \geq 0$ converge to the same lift and pitching moment coefficients at high angles-of-attack (Figs 3a, 3c)"

This appears to be approximately true up until stall, but then $\delta_c=10$ degrees shows higher lift and pitching moment coefficients

- Figure 4:

Why did you examine the effect of different fin shapes here with the cephalofoil removed, but not elsewhere in the paper? The fin shape seems to only be mentioned in one sentence, just above figure 4.

I am not sure how this is relevant to this study. If you just needed to show a no-cephalofoil model, could this not be overlaid onto figure 3 instead?

- Figures 6 and 7:

It is hard to see the data in regions with a lot of overlap. Perhaps lowering interior line weight, or using a solid, transparent fill instead of lines could make these figures easier to read.

Discussion

- This section is very short and needs improvement to interrelate how anatomical variables influence stability and speed limits. Comparison to published studies of shark swimming (live animal studies and other modeling studies) would be useful. Also, limitations inherent to the work and how they impact study findings needs to be discussed.

- Regarding unpowered glide, it is stated "...becomes equivalent to the case of $\gamma=\lambda_{cf}=0$, just rendered inconceivable"

I am not sure what you mean here. Is this supposed to say that this condition is unachievable, rather than inconceivable?

Decision letter (RSOS-200864.R0)

Dear Professor Iosilevskii,

The editors assigned to your paper ("Center-of-mass and minimal speed limits of the great hammerhead") have now received comments from reviewers. We would like you to revise your paper in accordance with the referee and Associate Editor suggestions which can be found below (not including confidential reports to the Editor). Please note this decision does not guarantee eventual acceptance.

Please submit a copy of your revised paper before 05-Aug-2020. Please note that the revision deadline will expire at 00.00am on this date. If we do not hear from you within this time then it will be assumed that the paper has been withdrawn. In exceptional circumstances, extensions may be possible if agreed with the Editorial Office in advance. We do not allow multiple rounds of revision so we urge you to make every effort to fully address all of the comments at this stage. If deemed necessary by the Editors, your manuscript will be sent back to one or more of the original reviewers for assessment. If the original reviewers are not available, we may invite new reviewers.

To revise your manuscript, log into <http://mc.manuscriptcentral.com/rsos> and enter your Author Centre, where you will find your manuscript title listed under "Manuscripts with

Decisions." Under "Actions," click on "Create a Revision." Your manuscript number has been appended to denote a revision. Revise your manuscript and upload a new version through your Author Centre.

- Data accessibility

If you wish to submit your supporting data or code to Dryad (<http://datadryad.org/>), or modify your current submission to dryad, please use the following link:
<http://datadryad.org/submit?journalID=RSOS&manu=RSOS-200864>

- Competing interests

- Authors' contributions

- Acknowledgements

- Funding statement

Kind regards,

Andrew Dunn

on behalf of Dr Jake Socha (Associate Editor) and Kevin Padian (Subject Editor)

Associate Editor's comments (Dr Jake Socha):

Associate Editor: 1

Comments to the Author:

The hydrodynamics of shark swimming is worthy of consideration, and both reviewers agree that this experimental study provides new insights on the topic. However, there are numerous technical concerns that need to be addressed in order for this manuscript to be considered further. Please address both reviewers' comments in the revised submission.

Associate Editor: 2

Comments to the Author:

(There are no comments.)

Comments to Author:

Reviewers' Comments to Author:

Reviewer: 1

Comments to the Author(s)

See the attached document.

Reviewer: 2

Comments to the Author(s)

Summary:

The author performs wind-tunnel testing of a 3D printed model of a hammerhead shark with varying body angle, cephalofoil angle, and pectoral fin angles in order to determine the minimal swimming speed, as well as the anterior and posterior center-of-mass limits that allow the shark to remain longitudinally balanced across the whole range of possible swimming speeds.

While there have been several studies on hydrodynamics of shark swimming, we still do not know how the physical design of specific shark (such as the hammerhead) can provide better maneuvering capability. Studies using modeling approaches as in this manuscript can help us to identify importance of specific morphological variables.

Before I can recommend the manuscript for publication, revisions are needed to improve readability in both the text and in the figures. Also, clarifications on some of the

statements/assumptions are needed and the discussion section needs to be significantly improved. Specific comments are provided below.

Abstract

- 'maximal margin between anterior and posterior center-of-mass limits turned out to be a diminutive $0.3(\beta)l$.

How diminutive is this? What length, or percentage of body length, is $0.3(\beta)l$ for a typical adult?

Introduction

- It would be helpful to provide an overview to the broad problem via discussion of previous studies. Background citations are strewn in places but there is no cohesive discussion of what we know and what remains unclear. Finally, the author uses the introduction to provide backing to more of the finer technical aspects that are introduced later in the study, making it difficult to read the manuscript from start to finish.

- Figure 1: Center of buoyancy location is given, and the distance between center-of-mass and center of buoyancy is stated to be exaggerated more than 10x.

Is this based on your limits for center-of-mass or based on previous work? If based on previous work, please indicate the location.

If L and D are supposed to be defined relative to the body orientation, the labels are slightly off (e.g., $D_{no\ cf}$ is not in line with cranio-caudal axis).

Finally, please define λ_{cf} in the caption.

- The use of gravity as a force, and weight as the difference between gravity and buoyancy can make the manuscript hard to follow at times.
- "There is no difference between swimming in water and flying in air as long as the respective Reynolds numbers are similar and no cavitation occurs."

If the Mach number approaches the transonic region, but the Reynolds number is similar then there is a difference. This is an extreme case, but so is cavitation.

In addition, the Strouhal number is important in unsteady flows that are generated by fin motion. However, static wind tunnel models as in this study do not provide true similarity with dynamic organism-swimming. Please address this.

Materials and methods

- "all fins had NACA 0015 profile"

Why was this profile chosen? Is it similar to hammerhead shark fins (if so cite papers that support such a claim)? Easier to manufacture?

- "Both forces were assumed to act at $(x_{cf}, 0, z_{cf})$, 0.685l posterior to the center-of-buoyancy" x , y , and z are not defined as a coordinate system. This could be added to figure 1.

- The definitions for the scaled swimming speed Fr_{hat} and scaled horizontal margin between the center of mass and center of buoyancy x_{hat} are not well explained in the main text of the paper, and rely heavily on the reader examining the appendix for explanation of their usage.

- Figure 2: please label components and show additional views for clarity (e.g., only 1 pectoral fin is shown in existing drawing).

Results

- Given that this was mainly an experimental study, it is important to include standard deviation or standard error in charts. It is possible to either show them in the form of error bars or shading of lines.

- "In general, both the lift and the pitching moment coefficients increase with the angle-of-attack and with the angle of the cephalofoil relative to the body, but the lift coefficient drops when the angle-of-attack exceeds 13 degrees"

Does this mean that the shark is dynamically unstable in the longitudinal direction, and has to constantly expend energy to keep swimming in a straight line instead of tumbling?

- "Consequently, all configurations with $\Delta_c \geq 0$ converge to the same lift and pitching moment coefficients at high angles-of-attack (Figs 3a, 3c)"

This appears to be approximately true up until stall, but then $\Delta_c = 10$ degrees shows higher lift and pitching moment coefficients

- Figure 4:

Why did you examine the effect of different fin shapes here with the cephalofoil removed, but not elsewhere in the paper? The fin shape seems to only be mentioned in one sentence, just above figure 4.

I am not sure how this is relevant to this study. If you just needed to show a no-cephalofoil model, could this not be overlaid onto figure 3 instead?

- Figures 6 and 7:

It is hard to see the data in regions with a lot of overlap. Perhaps lowering interior line weight, or using a solid, transparent fill instead of lines could make these figures easier to read.

Discussion

- This section is very short and needs improvement to interrelate how anatomical variables influence stability and speed limits. Comparison to published studies of shark swimming (live animal studies and other modeling studies) would be useful. Also, limitations inherent to the work and how they impact study findings needs to be discussed.

- Regarding unpowered glide, it is stated "...becomes equivalent to the case of $\gamma = \lambda_{cf} = 0$, just rendered inconceivable"

I am not sure what you mean here. Is this supposed to say that this condition is unachievable, rather than inconceivable?

Author's Response to Decision Letter for (RSOS-200864.R0)

See Appendix B.

RSOS-200864.R1 (Revision)

Review form: Reviewer 1

Is the manuscript scientifically sound in its present form?

Yes

Are the interpretations and conclusions justified by the results?

Yes

Is the language acceptable?

Yes

Do you have any ethical concerns with this paper?

No

Have you any concerns about statistical analyses in this paper?

Yes

Recommendation?

Accept as is

Comments to the Author(s)

The author has fully addressed my concerns. Now I recommend it for publication.

Review form: Reviewer 2

Is the manuscript scientifically sound in its present form?

Yes

Are the interpretations and conclusions justified by the results?

Yes

Is the language acceptable?

Yes

Do you have any ethical concerns with this paper?

No

Have you any concerns about statistical analyses in this paper?

Yes

Recommendation?

Accept as is

Comments to the Author(s)

The author has addressed my comments satisfactorily.

Decision letter (RSOS-200864.R1)

Dear Professor Iosilevskii,

It is a pleasure to accept your manuscript entitled "Center-of-mass and minimal speed limits of the great hammerhead" in its current form for publication in Royal Society Open Science. The comments of the reviewer(s) who reviewed your manuscript are included at the foot of this letter.

on behalf of Dr Jake Socha (Associate Editor) and Kevin Padian (Subject Editor)
openscience@royalsociety.org

Associate Editor Comments to Author (Dr Jake Socha):

Both reviewers were satisfied completely with the revisions to the manuscript, and no other changes are requested. Congratulations on acceptance of your contribution providing a new theoretical understanding of shark locomotion!

Reviewer comments to Author:

Reviewer: 1

Comments to the Author(s)

The author has fully addressed my concerns. Now I recommend it for publication.

Reviewer: 2

Comments to the Author(s)

The author has addressed my comments satisfactorily.

Appendix A

The topic of this paper is very interesting. Sharks in nature have a higher density than water. This leads to a sinking force on the body. Also, the density is not uniformly distributed, which results in a pitch torque due to the difference between the center of mass and the center of buoyance. These two factors require the sharks to use some strategies to balance the sinking force and the pitch torque during swimming. To find out the way, this work studies the position of the center of mass and the minimum swimming speed of the great hammerhead sharks using experiments and theoretical analysis. I feel the results of this paper are valuable. However, there are some points that need to be clarified before this manuscript can be recommended for publication.

1. All the analysis of this work is based on the assumption that the drag and lift of the shark body in swimming can be measured by a solid shark model under different steady-state flow conditions. However, the real fish swimming is an unsteady motion, and the flow is unsteady as well. It is not clear here if we could use this assumption because the hydrodynamic performance may be significantly different if the flow becomes unsteady. Also, how reliable is it by assuming the drag/lift of a solid body is close to a deforming (undulatory) body? So, more evidences are needed to support this assumption.
2. There are too many mathematic symbols in this manuscript, which makes the text challenging to follow. It would be better if the author could provide a nomenclature table, so that the readers do not have to go back and forth to find out their definitions.
3. The “center of pressure” was introduced suddenly by its mathematical definition (Eq. A20). Could the author add its physical meaning to the text? Besides, it would be clearer if it is labelled in figure 1.
4. Page 7, “lift of the caudal fin was assumed proportional to its thrust”. How reliable is this assumption? How does the angle of attack (α) affect λ_{cf} ? Are the sharks able to adjust the caudal fin lift while maintaining the same thrust via the flexibility of the tail?

Appendix B

Reviewer 1:

The topic of this paper is very interesting. Sharks in nature have a higher density than water. This leads to a sinking force on the body. Also, the density is not uniformly distributed, which results in a pitch torque due to the difference between the center of mass and the center of buoyance. These two factors require the sharks to use some strategies to balance the sinking force and the pitch torque during swimming. To find out the way, this work studies the position of the center of mass and the minimum swimming speed of the great hammerhead sharks using experiments and theoretical analysis. I feel the results of this paper are valuable. However, there are some points that need to be clarified before this manuscript can be recommended for publication.

1. All the analysis of this work is based on the assumption that the drag and lift of the shark body in swimming can be measured by a solid shark model under different steady-state flow conditions. However, the real fish swimming is an unsteady motion, and the flow is unsteady as well. It is not clear here if we could use this assumption because the hydrodynamic performance may be significantly different if the flow becomes unsteady. Also, how reliable is it by assuming the drag/lift of a solid body is close to a deforming (undulatory) body? So, more evidences are needed to support this assumption.

Drag and lift are not similar in this respect. Having assumed that most of the thrust is generated by the caudal fin, and not by the body, we can rationally assume that the lift generated by the body is also small. My recent study of hydrodynamics of a swimming sea snake has demonstrated that undulating snakes can generate lift of the same order of magnitude as thrust (Hydrodynamics of a twisting slender swimmer. *R. Soc. Open Sci.* 2020. 7: 200754, should appear next week). If thrust is small, so is the lift. Consequently, undulations of the body can affect lift only through left-right yawing motion of the pectoral fins and the cephalofoil, that can make the forces reconstructed from wind tunnel measurements to underestimate the true forces. The augmentation factor should be a fraction of the ratio between the mean-square forward velocity of the distal margin of the respective lifting surface, u , and the swimming speed, v . Assuming sinusoidal yawing motion with frequency f and angular amplitude θ_0 , $\langle u^2 \rangle / v^2 = 2(\pi f b \theta_0 / v)^2$, where b is the distance between that margin and the sagittal plane. By interpretation, $b\theta_0$ is its anterior-posterior displacement, whereas the ratio v/f is the stride length. The ratio of the two is invariably small, and so is $\langle u^2 \rangle / v^2$. A 3m shark, which probably had the distal margin of its pectoral fins situated ~ 0.45 m from the sagittal plane, was logged swimming at 0.8 m/s with tail-beat frequency of 0.4 HZ [9]. Assuming $\theta_0 \sim 0.1$ rad (<https://www.nature.com/articles/ncomms12289#Sec12>), $\langle u^2 \rangle / v^2 \sim 0.01$. I added a footnote at the end of Section 1.

Drag can be conceptually divided into a form drag (which is associated with separation of the boundary layer from the body), induced drag (which is associated with generation of lift), and friction drag (which is associated with shear stresses on the surface of the body). Friction drag can be affected only if there are local acceleration of the flow over the body. There is no evidence that this effect is significant. Form

drag can be affected only if undulations cause separation of the boundary layer. There is no evidence to that either. Induced drag can be affected only through variations in lift, which were rendered small in the preceding paragraph. I have avoided these arguments in the paper by *defining* drag as the straight-body drag and (tacitly) attributing possible variations in drag to thrust, which was (rightly) assumed to be contributed mainly by the caudal fin.

2. There are too many mathematic symbols in this manuscript, which makes the text challenging to follow. It would be better if the author could provide a nomenclature table, so that the readers do not have to go back and forth to find out their definitions.

Nomenclature section has been added

3. The “center of pressure” was introduced suddenly by its mathematical definition (Eq. A20). Could the author add its physical meaning to the text? Besides, it would be clearer if it is labelled in figure 1.

I could not add it on Fig.1 because it would have made it too crowded. However, I did move the definition of the center of pressure from the appendix into Section 2.4 of Materials and Methods.

4. Page 7, “lift of the caudal fin was assumed proportional to its thrust”. How reliable is this assumption? How does the angle of attack (α) affect λ_{cf} ? Are the sharks able to adjust the caudal fin lift while maintaining the same thrust via the flexibility of the tail?

One can always use the ratio between two free parameters instead of one of them. No harm is done as long as the ratio between lift and thrust of the caudal fin is not fixed. It was not fixed in the paper, and one of central conclusions of the paper is that it cannot be fixed – if fixed at zero, the shark will have an impossibly small mass-buoyancy margin; if fixed different from zero, the shark will tumble. Hence a shark must have control over the lift to thrust ratio of its caudal fin, moreover, it must have the ability to generate lift with negative thrust (drag). I am afraid that this point was not clear enough in the original version and it is now included both in Discussion and Conclusions (Sections 4 and 5).

Reviewer 2:

The author performs wind-tunnel testing of a 3D printed model of a hammerhead shark with varying body angle, cephalofoil angle, and pectoral fin angles in order to determine the minimal swimming speed, as well as the anterior and posterior center-of-mass limits that allow the shark to remain longitudinally balanced across the whole range of possible swimming speeds. While there have been several studies on hydrodynamics of shark swimming, we still do not know how the physical design of specific shark (such as the hammerhead) can provide better maneuvering capability. Studies using modeling approaches as in this manuscript can help us to identify importance of specific morphological variables. Before I can recommend the manuscript for publication, revisions are needed to improve readability in both the text and in the figures. Also, clarifications on some of the statements/assumptions are needed and the discussion section needs to be significantly improved. Specific comments are provided below.

Abstract

- 'maximal margin between anterior and posterior center-of-mass limits turned out to be a diminutive $0.3(\beta)$ l. How diminutive is this? What length, or percentage of body length, is $0.3(\beta)$ l for a typical adult?

β is a few hundredths with most elasmobranchs, and hammerheads are not an exception. 0.3β is about a hundredth. I have changed the Abstract.

Introduction

- It would be helpful to provide an overview to the broad problem via discussion of previous studies. Background citations are strewn in places but there is no cohesive discussion of what we know and what remains unclear. Finally, the author uses the introduction to provide backing to more of the finer technical aspects that are introduced later in the study, making it difficult to read the manuscript from start to finish.

As far as I am aware of, there were no studies on center-of-mass limits. I have added a comparison with the existing studies of the roles of pectoral fins and heterocercal tail in Conclusion (Section 5)

- Figure 1: Center of buoyancy location is given, and the distance between center-of-mass and center of buoyancy is stated to be exaggerated more than 10x. Is this based on your limits for center-of-mass or based on previous work? If based on previous work, please indicate the location. If L and D are supposed to be defined relative to the body orientation, the labels are slightly off (e.g., $D_{no\ cf}$ is not in line with cranio-caudal axis).

As far as I am aware of, nobody talked about the distance between the two centers before. The buoyancy gravity couple is the product of the body weight out of water and the distance between the centers of mass and buoyancy. This couple needs to be counteract by hydrodynamic pitching moment, which is the product between the hydrodynamic lift and the distance between the centers of pressure and buoyancy. As hydrodynamic lift is the submerged weight, a few hundredths of the weight out of water, the distance between the centers of gravity and buoyancy cannot exceed a few hundredths of the distance between the centers of pressure and buoyancy. Hence the 10x magnification. I removed the '10x' from the caption.

L and D are, by definition, the components of hydrodynamic force relative to the swimming direction, the former perpendicular to it, the latter along it. The figure reflects it.

Finally, please define λ_{cf} in the caption.

Done.

- The use of gravity as a force, and weight as the difference between gravity and buoyancy can make the manuscript hard to follow at times.

It was never meant to be the difference between gravity and buoyancy (i.e. the submerged weight), but I see from where the confusion could arise. I changed the notation.

- "There is no difference between swimming in water and flying in air as long as the respective Reynolds numbers are similar and no cavitation occurs." If the Mach number approaches the transonic region, but the Reynolds number is similar then there is a difference. This is an extreme case, but so is cavitation.

In spite of my teaching a course on compressible aerodynamics while writing this paper, compressibility effects never crossed my mind in this context. You are certainly right. I've changed the write-up.

In addition, the Strouhal number is important in unsteady flows that are generated by fin motion. However, static wind tunnel models as in this study do not provide true similarity with dynamic organism-swimming. Please address this.

Unsteady effects have little to do with the lift of the shark without the caudal fin. A change in lift can come only from the yawing motion of the pectoral fins (and, to some extent, the cephalofoil), which can make the effective velocity of these surfaces slightly larger than the swimming velocity. The effect is small. Please see a detailed response to the first referee. I added a footnote in Section 1.

The lift of the caudal fin does, of course, depend on the tail-beat frequency, but this dependence is irrelevant to the line of arguments laid in the paper - all forces are period-averaged.

Materials and methods

- "all fins had NACA 0015 profile" Why was this profile chosen? Is it similar to hammerhead shark fins (if so cite papers that support such a claim)? Easier to manufacture?

There is no systematic data on profiles of the pectoral fins and the cephalofoil of sharks in general, and the great hammerhead in particular. At chord-based Reynolds-numbers between 100 and 150 thousands, which are relevant to this study, a ubiquitous NACA 0015 performs practically as well as any symmetrical airfoil with thickness ratio between 9 and 18 percent [12,13,14] and is thick enough to make 3D printing in FullCure720 worries-free. I added a footnote in Material and Methods, Section 2.1.

- "Both forces were assumed to act at $(x_{cf}, 0, z_{cf})$, 0.685l posterior to the center-of-buoyancy". x , y , and z are not defined as a coordinate system. This could be added to figure 1.

I have moved the description of the three reference frames from the appendix into Materials and methods (Section 2.2.). Two of these systems are shown now on Fig. 1.

- The definitions for the scaled swimming speed \hat{Fr} and scaled horizontal margin between the center of mass and center of buoyancy \hat{x} are not well explained in the main text of the paper, and rely heavily on the reader examining the appendix for explanation of their usage.

I am not sure what to do with this comment. Derivations are bulky and hardly inviting, and therefore they were moved to an appendix. They are not needed to understand the results. I hoped that an interested reader will have the patience to follow the appendix. I repeated equations (A15), (A16) as

well as the definition of the center of pressure in the text. These are now equations (2), (3) and (4) in Materials and Methods. I hope that it is better.

- Figure 2: please label components and show additional views for clarity (e.g., only 1 pectoral fin is shown in existing drawing).

I am afraid I cannot do much here. This is the best view I can provide. The drawings in the supplementary should answer all the questions about the model. The model itself can be downloaded from Dryad.

Results

- Given that this was mainly an experimental study, it is important to include standard deviation or standard error in charts. It is possible to either show them in the form of error bars or shading of lines.

As opposed to biological experiments, statistical error in static wind tunnel experiments is next to nonexistent. There is nothing to show. Small scatter starts post-stall, but it is mostly irrelevant. The last sentence of Section 2.3 in Materials and Methods notes it.

- "In general, both the lift and the pitching moment coefficients increase with the angle-of-attack and with the angle of the cephalofoil relative to the body, but the lift coefficient drops when the angle-of-attack exceeds 13 degrees" Does this mean that the shark is dynamically unstable in the longitudinal direction, and has to constantly expend energy to keep swimming in a straight line instead of tumbling?

Absolutely. Nonetheless, I am not sure that it spends much energy on stabilization - slight adjustment of the tail lift and of the pectoral fins will do the job. You do not spend much energy on balancing a bike. I did not touch static stability issues in this paper, because it is largely irrelevant when near neutral buoyancy. My student mapped the dynamic modes. The fastest diverging mode still has time to double amplitude of a few tail-beats.

- "Consequently, all configurations with $\delta_c \geq 0$ converge to the same lift and pitching moment coefficients at high angles-of-attack (Figs 3a, 3c)" This appears to be approximately true up until stall, but then $\delta_c = 10$ degrees shows higher lift and pitching moment coefficients

Post-stall results are hardly relevant because of the huge drag that associated with them (Fig. 3b) – in fact, it becomes of the same order of magnitude as lift. This is not a posture for locomotion.

- Figure 4: Why did you examine the effect of different fin shapes here with the cephalofoil removed, but not elsewhere in the paper? The fin shape seems to only be mentioned in one sentence, just above figure 4. I am not sure how this is relevant to this study. If you just needed to show a no-cephalofoil model, could this not be overlaid onto figure 3 instead?

I admit going astray here, because neither the removal of the cephalofoil, nor the shape (or the removal) of the pectoral fins are directly related to the main narrative of the paper, which is the center-of-mass and speed limits. They are related, however, to the broader narrative of this study, which is the

longitudinal balance of a swimming shark, and I thought that it would be a waste not to share them. Figure 3 is already almost too crowded. Figure 4 came to concentrate all non-standard configurations of the shark – actually, none of the configurations shown there pertain to a hammerhead.

- Figures 6 and 7: It is hard to see the data in regions with a lot of overlap. Perhaps lowering interior line weight, or using a solid, transparent fill instead of lines could make these figures easier to read.

As far as I know, Matlab does not allow a transparent fill. I reduced the number of connecting lines and their width. I hope that the figures read better now.

Discussion

- This section is very short and needs improvement to interrelate how anatomical variables influence stability and speed limits. Comparison to published studies of shark swimming (live animal studies and other modeling studies) would be useful. Also, limitations inherent to the work and how they impact study findings needs to be discussed.

The section was significantly extended. It is now Section4. Concluding section was added as well.

- Regarding unpowered glide, it is stated “....becomes equivalent to the case of $\gamma=\lambda_{cf}=0$, just rendered inconceivable” I am not sure what you mean here. Is this supposed to say that this condition is unachievable, rather than inconceivable?

I changed the write-up.